# Real-time holographic lensless micro-endoscopy through flexible fibers via fiber bundle distal holography

### Noam Badt [1] & Ori Katz[1] ✉

Fiber-based micro-endoscopes are a critically important tool for minimally-invasive deep-tissue imaging. However, current micro-endoscopes cannot perform three-dimensional imaging through dynamically-bent fibers without the use of bulky optical elements such as lenses and scanners at the distal end, increasing the footprint and tissue-damage. Great efforts have been invested in developing approaches that avoid distal bulky optical elements. However, the fundamental barrier of dynamic optical wavefront-distortions in propagation through flexible fibers limits current approaches to nearly-static or non-flexible fibers. Here, we present an approach that allows holographic, bend-insensitive, coherence-gated, micro-endoscopic imaging using commercially available multi-core fibers (MCFs). We achieve this by adding a partially-reflecting mirror to the distal fiber-tip, allowing to perform low-coherence full-field phase-shifting holography. We demonstrate widefield diffraction-limited reflection imaging of amplitude and phase targets through dynamically bent fibers at video-rate. Our approach holds potential for label-free investigations of dynamic samples.

Flexible optical micro-endoscopes are an important tool for a wide variety of applications, from clinical procedures to biomedical investigations. In such applications micron-scale structures such as single neurons are imaged[1,2] or optically excited[3] at depths beyond the reach of conventional microscopes. Developing a flexible, video-rate, label-free micro-endoscope with a minimal footprint is thus a sought-after goal for minimally invasive deep-tissue imaging.

In the quest for this goal, various micro-endoscopes have been developed in recent years. One set of solutions consists of micro-endoscopes that are based on single-mode fibers. While bend-insensitive, these require distal optical elements such as scanners and lenses[1], or spectral dispersers[4,5] to produce an image. Such distal elements significantly enlarge the endoscope diameter, increasing tissue damage, and consequently limiting its use for minimally-invasive deep-tissue imaging.

Another set of solutions utilizes imaging fiber bundles, also known as multi-core fibers (MCFs). These consist of thousands of cores closely packed together. Conventionally, each core functions as a single pixel, either by direct contact with the target or by the addition of a GRIN lens at the distal end[1,2]. While straightforward to use, conventional bundle-based endoscopes suffer from limited resolution, pixelation, and a small and fixed working distance with poor axial sectioning capability. Even though axial-sectioning can be improved by using temporal coherence-gating[6,7] or confocal or other structured illumination[8,9], many of the inherent limitations remain. These include pixelation artefacts, limited resolution, and a fixed working distance. Moreover, strong spurious reflections from the fiber tips limit the applicability of such approaches for label-free imaging.

An emerging set of approaches for lensless endoscopes that go beyond many of the conventional limitations are based on the measurement of the fiber transmission-matrix (TM), with or without an additional distal mask or diffuser. This allows compensation of the complex coherent or incoherent transfer-function of these multi-mode systems, either digitally[10–16] or via wavefront-shaping[17–24]. However, solutions that are based on a single TM measurement without the addition of a distal element are limited to non-flexible fibers, since fiber

---

[1]Department of Applied Physics, The Hebrew University of Jerusalem, Jerusalem 9190401, Israel. ✉e-mail: orik@mail.huji.ac.il

bending strongly distorts the TM. The TM may be estimated using a large number of measurements and complex computation, but do not allow video-rate imaging or imaging through dynamically bent fibers[25–27]. Alternative, speckle-based incoherent imaging approaches[11–14], suffer from low contrast of the acquired images. MCFs with bend-insensitive inter-core phase relations have been fabricated[28]. However, these rely on a relatively small number of single-mode cores, suffering from low collection efficiency and fill-factor.

Related to the TM-based approaches, Czarske et al.[29–31] have recently demonstrated in-situ measurement of the fiber phase-distortions by adding a partially-reflecting mirror to the MCF distal-facet, and correction using a spatial light modulator (SLM). However, as imaging is performed by raster scanning a phase-corrected focus, correction of dynamic distortions is challenging, and fluorescence labeling is required.

Label-free holographic approaches to endoscopy that utilize MCFs as intensity-only image-guides have been put forward over the years[32–34]. However, these rely on additional fibers for illumination, and suffer from poor axial-sectioning, and limited field of view (FoV) or coherent background due to twin image artefacts.

Here we present Fiber Bundle Distal Holography (FiDHo): a simple approach that allows holographic, calibration-free, bend-insensitive, coherence-gated, micro-endoscopic imaging utilizing commercially available MCFs. We achieve this by adding a partially reflecting mirror to the distal fiber tip (Fig. 1), which allows us to perform low-coherence full-field phase-shifting holography at video rate, with diffraction-limited resolution, where the coherent illumination is provided through a single MCF core.

## Results
### Principle

The optical setup for realizing our approach is schematically depicted in Fig. 1. It is based on the excitation of a single MCF core by two delayed replicas of a low coherence illuminating beam (Fig. 1A). The two beams co-propagate in the same core along the MCF. At the MCF distal end, the reflection of the first arriving beam from the target interferes with the reflection of the second ("reference") beam from the distal partially-reflecting mirror (Fig. 1B). The interference intensity

pattern is collected by the MCF cores (Fig. 1C) and relayed back to the proximal end where it is imaged on a camera (Fig. 1A).

The object complex field at the distal facet is reconstructed from $N \geq 3$ camera frames via low-coherence phase-shifting holography, performed by controlling the phase of the reference beam (Fig. 1A). The use of a low-coherence source allows effective rejection of all spurious reflections by setting the time delay between the two excitation beams, $\tau$, to match the object distance from the mirror, $z_0$: $\tau = 2z_o/c$. Both the time delay, $\tau$, and the phase-shifting are performed using the same interferometer that produces the two excitation beams.

The recorded hologram at the n-th phase shift, $I_n$, is the interference pattern from the diffracted object field, $E_o$, and the known, fixed spherical reference field, $E_r$, that is reflected from the distal mirror: $I_n(x,y) = |E_o(x,y) + E_r(x,y)e^{i\varphi_n}|^2$, where $\varphi_n = \frac{n}{N}2\pi$ is the phase added to the reference field. Importantly, the recorded interference intensity pattern is insensitive to fiber bending since both the reference and illumination beams propagate through the same fiber core, and the MCF faithfully relays the holograms intensity images. Finally, the 3D object field is digitally reconstructed from the recorded field at the distal facet via back-propagation, as explained below.

### Reconstruction process

The reflected object field at the fiber distal tip, $E_o$, is reconstructed from $N \geq 3$ camera frames, $I_n$ (Fig. 2A), by phase-shifting interferometry, followed by normalization with the conjugate of the known reference field, $E_r^*$ (Fig. 2B):

$$E_o(x,y) = \frac{1}{E_r^*(x,y)} \sum_{n=1}^{N} I_n(x,y)e^{i\varphi_n} \qquad (1)$$

where $E_r^*$ is approximated as a Gaussian beam from the theoretical first mode of a single fiber core.

3D reconstruction is then performed by back-propagation of the recorded field (Fig. 2D), $E_o$, to any desired distance from the fiber facet, $z_{prop}$, and sequential normalization by the expected illumination field

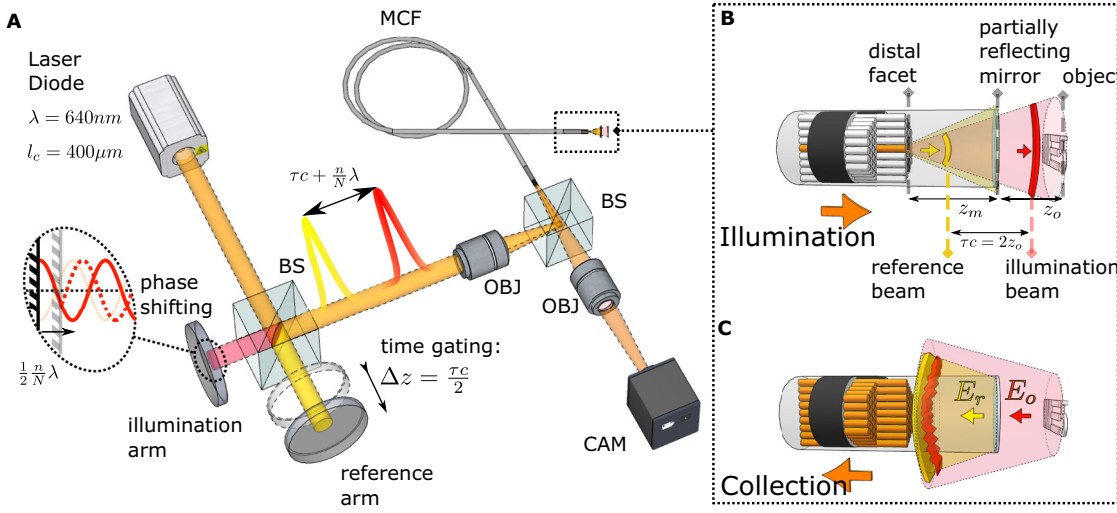

**Fig. 1 | Setup and principle of distal holographic endoscopy. A** A short coherence laser (orange) is split into two delayed replicas: an illumination beam and a reference beam (red and yellow, respectively) that are coupled into a single core of a multicore fiber (MCF). **B** At the distal MCF facet, the illumination beam and the reference beam are reflected from the target object and a partially reflecting distal mirror, respectively. **C** The intensity pattern of the interference between the reflected beams is relayed by the MCF to the proximal side, where it is imaged on a

camera (**A**). Due to coherence gating, setting the relative time delay between the two arms to match the object-mirror distance, $\tau = 2Z_o/c$, results in measured interference only between the reflected illumination from the object and the reflected reference from the mirror. The object complex field is retrieved from $N \geq 3$ intensity-only proximal images, by phase-shifting the illumination arm (**A**, inset). See Fig. S1 for the full setup.

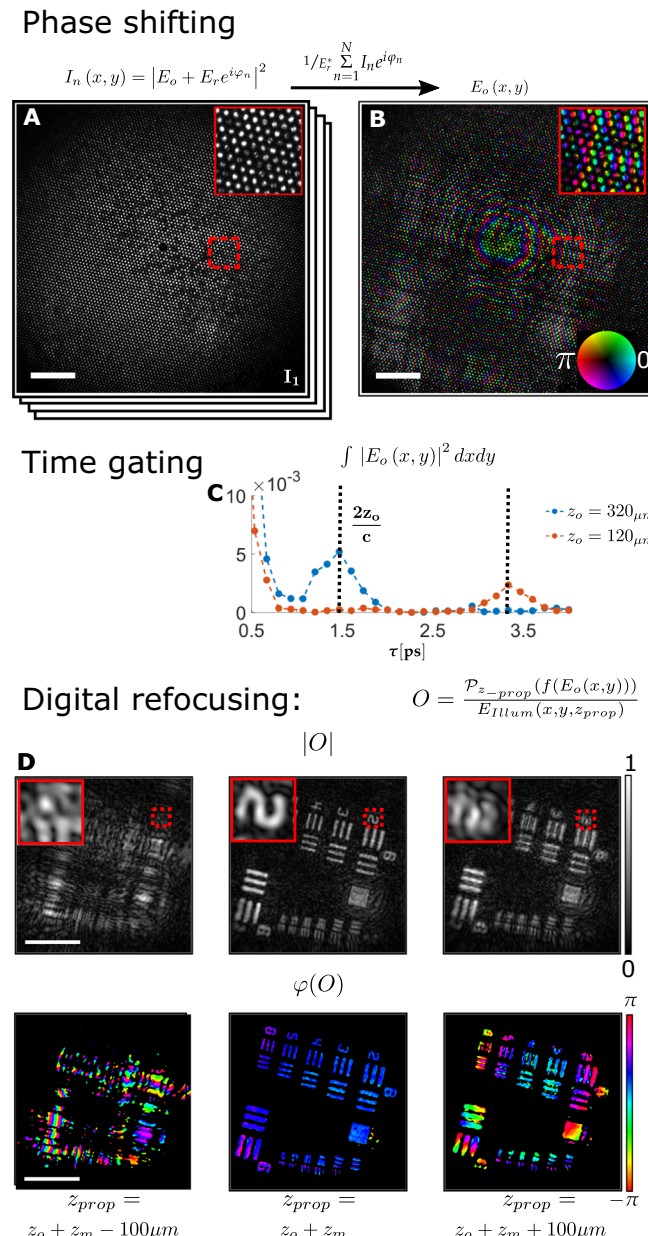

## Phase shifting

$$I_n(x,y) = |E_o + E_r e^{i\varphi_n}|^2 \xrightarrow{1/E_r^* \sum_{n=1}^{N} I_n e^{i\varphi_n}} E_o(x,y)$$

## Time gating

$$\int |E_o(x,y)|^2 dxdy$$

## Digital refocusing:

$$O = \frac{\mathcal{P}_{z-prop}(f(E_o(x,y)))}{E_{Illum}(x,y,z_{prop})}$$

**Fig. 2 | Experimental holographic imaging of a resolution target. A** $N$ phase-shifted proximal intensity images $I_n(x,y)$ are used to retrieve the reflected object field at the distal facet, $E_o(x,y)$ (**B**). **C** Scanning the time delay between the reference and illumination arms reveals a peak of the total retrieved field energy at the target distance, verified by placing a reflective target at two different distances (orange, blue); **D** Digital refocusing is achieved by back-propagating the measured distal field $E_o(x,y)$. The back-propagated field reveals the reflective USAF target in focus (top center) and with a flat phase (bottom center) at the correct distance ($z_{prop} = z_o + z_m$, $z_o = 340$ μm), after normalizing by the illuminating field. At other back-propagation distances (left and right) the target is out-of-focus (Insets, zoom-in on dashed rectangles). Scale bars: 100 μm.

amplitude and phase $E_{illum}(x,y,z_{prop})$:

$$O(x,y,z_{prop}) = \frac{\mathcal{P}_{-z_{prop}}(E_o(x,y))}{E_{illum}(x,y,z_{prop})} \qquad (2)$$

Where $O$ is the reconstructed object complex reflection coefficient, $\mathcal{P}_{-z_{prop}}$ is the angular spectrum propagation operator by a distance $-z_{prop}$, and $E_{illum}$ is approximated as a Gaussian beam from the

theoretical first mode of a single fiber core. A digitally refocused sharp image of the object amplitude and phase is obtained (Fig. 2D) at the propagation distance $z_{prop} = z_o + z_m$, where $z_o$ is the object-mirror distance and $z_m$ is the mirror-fiber distance (Fig. 1B). The object distance can also be found from the low-coherence holograms by scanning the time delay, $\tau$, and plotting the total energy of the reconstructed field at each time delay (Fig. 2C). The minimal working distance is thus a function of the coherence-length of the laser used, and the object reflectivity. The propagator $\mathcal{P}_{-z_{prop}}$ can be implemented to take into account changes in refractive index, as well as incorporate computational aberration correction[35].

As can be observed in (Fig. 3A–H), unlike conventional MCF-based endoscopes, the holographic reconstruction is un-pixelated, since the MCF pixelation (i.e. spatial sampling) occurs at a different axial plane. The MCF pixelation at the fiber facet, which may introduce ghosting due to aliasing in k-space, is removed by digital filtering, and k-space aliasing is suppressed by the illumination and detection geometry set by the reference mirror distance (see "Digital filtering of MCF pixelation" in Methods and Supplementary section S3).

Beyond the 3D holographic amplitude and phase label-free reconstruction, FiDHo has several additional merits: it is insensitive to dynamic random phase distortions introduced by fiber bending, the full-field reconstruction requires only 3 frames, allowing video-rate imaging, and the low-coherence gating improves axial sectioning. In the next sections we experimentally demonstrate each of these merits.

## Resolution and field-of-view

To evaluate the system resolution and FoV, we performed several sets of experiments using reflective targets placed at different distances from the fiber (Fig. 3). Unlike conventional MCF-based microendoscopy, where each fiber core serves as one imaging pixel and the resolution is limited by the core-to-core pitch (Fig. 3A), in FiDHo the resolution is diffraction-limited, and the images are unpixelated (Fig. 3C, D). Specifically, FiDHo easily resolves group 7 element 4 of a USAF resolution target, signifying resolution better than 2.7 μm (Fig. 3G). More than two folds improvement over conventional contact mode (Fig. 3E). In addition to imaging a USAF resolution target at several distances (Fig. 3C, D), a precise quantification of the resolution and FoV was performed by imaging a knife-edge mirror and a large mirror respectively (Fig. 3I, J). Interestingly, and as expected from the theoretical analysis (see Supp Section S1), The resolution and FoV are independent of imaging distance for distances of $z_{obj} < nD_{fiber}/2NA - z_m$, where $D_{fiber}$ is the MCF diameter, n is the index of refraction of the medium, and $NA$ is the effective numerical aperture of the FiDHo imaging system. In the presented experimental system: $D_{fiber} = 600$ μm, $z_m = 2$ mm, $n = 1.51$, $NA \approx 0.15$, gives a constant resolution for $z_{obj} < 1$ mm $z_{obj} \gtrsim 1$mm. The resolution is diffraction-limited by the numerical aperture (NA) of the recorded holograms, which is dictated by the NA of the fundamental mode used for illumination and detection, and by the digital low-pass filtering of the holograms at $f_{cutoff} = 1/2p$ that is performed to filter artefacts caused by pixelation (sampling) of the ordered MCF cores, where $p$ is the core-to-core pitch (see Methods and Supplementary Section S3).

The FoV is half the fiber diameter. For our experimental parameters ($z_m \approx 2$ mm, $D_{fiber} \approx 600$ μm), the measured resolution and FoV (250 μm, full width at half max) indicate an effective $NA$ of $NA_{eff} \sim 0.15$, and a space-bandwidth product of ~5300. In line with the fiber parameters (Schott 153,385) and the interpolation performed (see "Digital filtering MCF pixelation" in Methods section).

The axial resolution of FidHO is determined by two factors: the axial sectioning due to coherence gating, and axial resolution due to the numerical aperture of the imaging system. First, similar to OCT, the coherence gating yields an axial sectioning with an axial resolution dictated by the coherence length of the laser used, $l_c$. In our

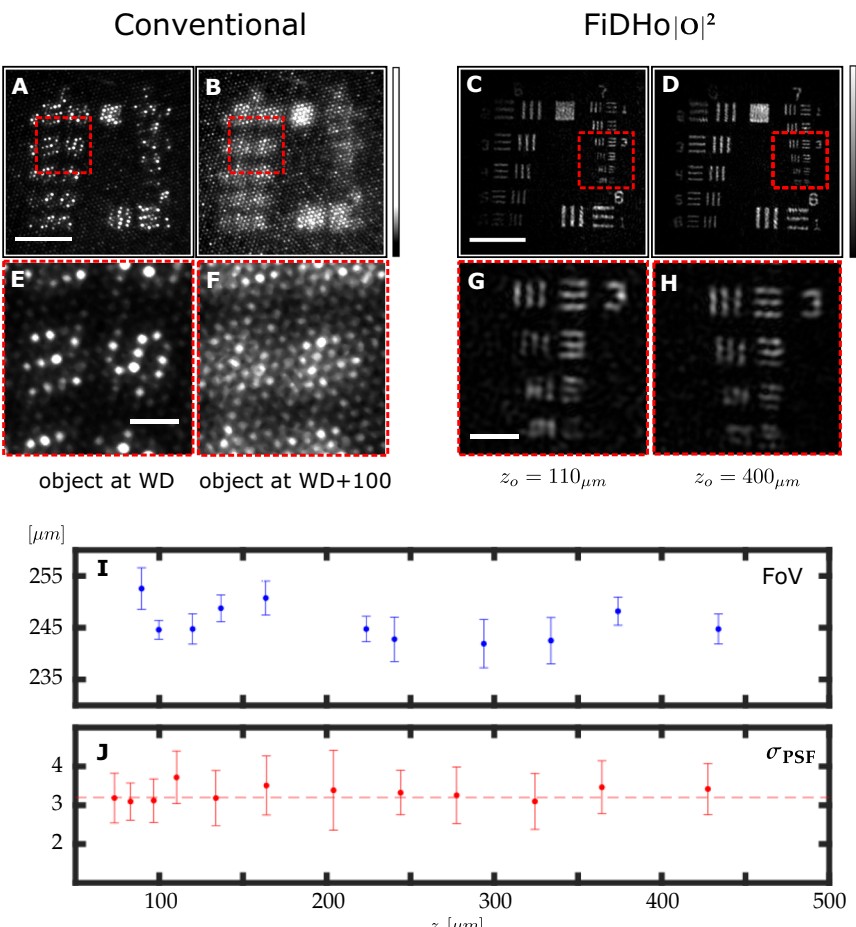

**Fig. 3 | Resolution and field-of-view characterization.** Unlike conventional MCF-based microendoscopy, where each fiber core serves as one imaging pixel, the resolution is limited by the core-to-core pitch (**A**), and the object (a USAF 1951 target) is pixelated and in focus only at a specific working-distance (**A**, **B**). In FiDHo **C**, **D**, the object is unpixellated and digitally focused at any distance, resolving features with a resolution that exceeds the fiber pixel pitch (**E–H**, zoom-in on dashed rectangles in **A–D**). **I** The system FoV, as measured by $1/e^2$ of the reconstructed intensity of a flat mirror. Blue dots and error-bars indicate the mean and standard deviation over $N = 6$ different crossections across the FoV at 30° increments, respectively (**J**) Imaging resolution as a function of object distance, as retrieved from a knife-edge measurement. Red dots and error-bars indicate the mean and standard deviation over the center $N = 100$ rows of the reconstructed image, respectively (see Supp. Section S1). Scale bars: **A–D** 100 μm, **E–H** 25 μm.

experiments $l_c = 400$ μm. However, a shorter coherence-length source can be used to improve the axial sectioning resolution. An experimental measurement of the coherence-gating axial sectioning in our experiments is presented in Supp. Fig. S7.

As in other holographic imaging approaches, in addition to the coherence-gating axial resolution, the imaging axial resolution that originates from the NA of the recorded holograms is dependent on the imaged object: for a point-reflector, the axial resolution would be $\delta z \approx \frac{\lambda}{NA^2} \approx 30$ μm. For a flat-phased object of diameter $d_{obj}$, $\delta z \approx \frac{d_{obj}^2 + (\frac{\lambda}{NA})^2}{\lambda}$ . The axial and lateral resolutions can be improved by advanced computational reconstruction schemes[36] (see Discussion).

## Phase-contrast imaging

The holographic nature of FiDHo has inherently phase-contrast imaging capability, important for the study of a wide variety of biological targets that presents very low reflection or absorption contrast and is not available in conventional endoscopes[37].

To demonstrate phase-contrast imaging, we imaged human cheek cells placed on a glass slide immersed in water. While the cells are not visible in the reconstructed amplitude (Fig. 4A), the reconstructed phase clearly reveals the cells (Fig. 4B), as is validated by a transmission image of the sample recorded by a reference distal camera (Fig. 4C, D).

## Video-rate and sensitivity to bending

An important advantage of FiDHo is its high frame rate. A single FiDHo frame requires three camera frames and thus is limited mainly by the camera framerate. Supplementary Movie S1 demonstrates the reconstruction of a moving target at 50 FPS. Individual frames are shown in Fig. 5A.

An additional major advantage of FiDHo is its inherent low sensitivity to fiber bending in both illumination and detection, an important requirement for in-vivo and freely-behaving animal studies[3,38]. In the illumination path, the illumination has inherently low sensitivity to bending since both the illumination and reference beams travel through the same core and mode. In the collection path, since only intensity is collected through the MCF cores, the detection is essentially insensitive to fiber bending. The inherently low sensitivity to bending together with the high imaging speed enable imaging through dynamically bent fibers, as we demonstrate in Supplementary Movie S2 and Fig. 5B. As can be observed in Supplementary Movie S2, while fast dynamic bending within the three phase-shifting frames may impact the imaging quality, the approach is insensitive to fiber orientation.

Each FiDHo reconstructed image requires at least 3 phase-shifting camera frames. In the Supplementary movies, each 3 consecutive camera frames are used to reconstruct one image. Therefore, while the images are reconstructed at the camera frame-rate of 50 frames-per-

second, two out of three camera frames are shared between consecutive reconstructions, i.e., there exist 16.6 unique reconstructed frames per second. In our experiments the imaging speed was limited by the sCMOS camera frame-rate. Improved imaging speeds can be obtained by using cameras with higher frame-rates and by replacing the piezo-stage by a faster phase-shifting mechanism, e.g. an acousto-optic modulator.

## Axially-sectioned imaging

The low-coherence time gating of FiDHo allows axial sectioning of 3D targets[39]. To demonstrate this, we have performed several experiments whose results are presented in Fig. 6. The experiments include

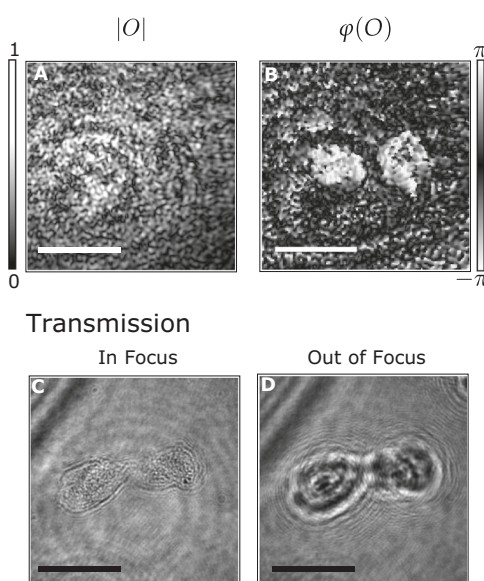

$|O|$                    $\varphi(O)$

### Transmission
In Focus                Out of Focus

**Fig. 4 | Phase-contrast imaging (A, B).** Reconstructed image of two cheek cells, placed in water on a microscope slide. The cells, which are not visible in the reconstructed field amplitude (**A**), are clearly visible in the reconstructed phase (**B**). **C, D** Reference In-focus (**C**) and out-of-focus (**D**) widefield transmission microscope images of the cells. Scale bars- 80 μm.

imaging of a sample composed of two stacked USAF targets with an axial separation of 300 μm (Fig. 6A) and a tilted USAF reflectivity target (Fig. 6B). In both experiments, the low-coherence gating together with the holographic reconstruction allows axial-sectioning, depth measurement, and digitally refocused reconstruction of all sample planes. While the transverse resolution is diffraction-limited, the axial resolution is set by the coherence length of the illumination source.

## Discussion

We have demonstrated an approach for lensless endoscopy that possess a unique set of advantages, not jointly attainable with current approaches. These include video-rate imaging with a transverse diffraction-limited resolution, axially-sectioned label-free imaging, in a calibration-free, bend-insensitive system, without any moving distal elements. Importantly, all these advantages are obtained in a simple system that employs commercially available MCFs, and with a straightforward non-iterative computational reconstruction.

For optimal imaging performance the parameters of the MCF, distal-mirror, and powers ratio of the illumination- and reference-arm should be set according to several considerations. These considerations are analyzed in detail in Supplementary Section S2. To summarize, at close imaging distances the fiber diameter, $D$, dictates the FoV and maximal depth for diffraction-limited resolution (Fig. 3), and should be as large as allowed by the selected application. The $NA$ of MCF cores dictate the imaging resolution.

Bend sensitivity can be minimized by choosing an MCF with low inter-core crosstalk and single-mode propagation in each core. This is since multimode propagation or crosstalk between cores will result in a complex and bend-sensitive illumination- and reference-fields when a fiber is bent. Nonetheless, by careful excitation and temporal-gating of the fundamental mode we were able to demonstrate low sensitivity to bending using commercial MCFs with cores that support several transverse modes. An in-depth analysis and quantification of the effects of high-order fiber modes is presented in Supplementary Section S4.

The distal mirror distance, $z_m$, should be large enough such that the reference-wave phase is approximately constant over each individual core, yielding: $z_m \gg \frac{Dd}{2\lambda}$, where $d$ is a single core diameter. Finally, the optimal mirror reflectivity is dependent on the target reflectivity and is approximately $2\% - 14\%$ for a reflected power

### 50FPS

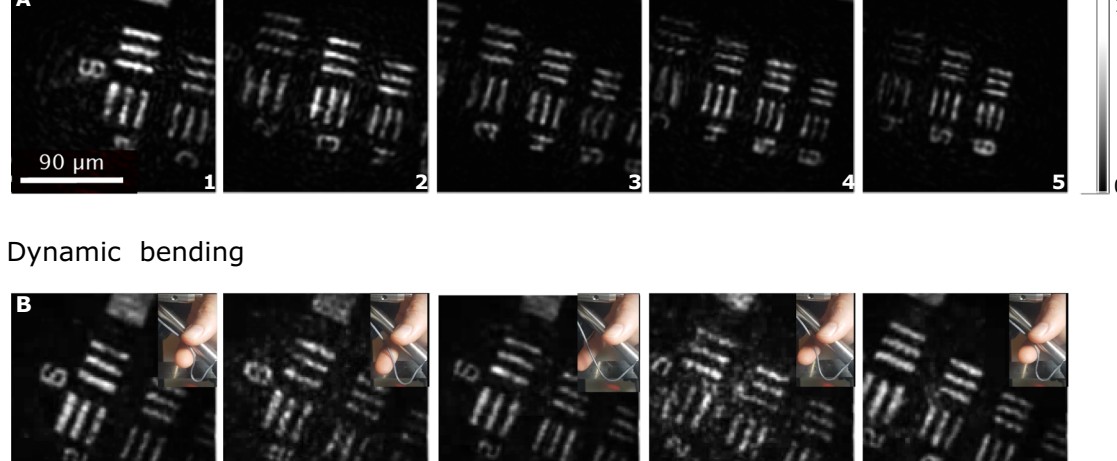

90 μm

### Dynamic bending

**Fig. 5 | Dynamic imaging at video-rate. A** Selected frames from a real-time video at 50 frames-per-second (FPS) of a moving resolution target (see Supplementary Movie S1). **B** same as (**A**) when the fiber is dynamically bent, showing the

insensitivity to fiber orientation, and low sensitivity to bending (see Supplementary Movie S2).

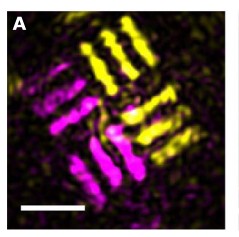
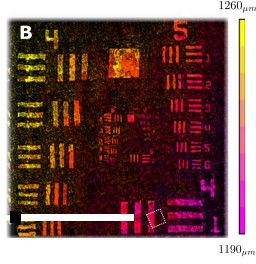

**Fig. 6 | Imaging three-dimensional objects. A** A 3D image of a target composed of two stacked resolution targets with a spacing of ∼ 300 μm reconstructed by superposing two reconstructed images acquired with two appropriate time-gates delays, $\tau = 2z_o/c$, with $z_o = 1030$ μm (pink) and $z_o = 1320$ μm (yellow). **B** A tilted USAF reflectivity target, reconstructed by stitching 20 × 20 sub-images, each with the fiber field of view given by the dashed square. The varying depth of each sub-image is retrieved from the time-gating delay scan. The target was placed at an angle w.r.t the distal tip to demonstrate 3D imaging capabilities. Scale bars: **A** - 50 μm **B** - 500 μm.

fraction of $0.1\% - 4\%$ from the target. While in our proof-of-principle experiments the distal mirror was held separately from the fiber, it can be incorporated in a miniature tip composed out of a cylinder glass spacer (see Supplementary Section S5 and Supp. Fig S9).

Our approach assumes that both the object illumination and the reference beam illumination originate from the fundamental mode of the fiber bundle core. While each core supports also higher-order modes, their measured arrival times are temporally separated by delays that are introduced at the illumination path by the fiber modal group velocity dispersion (GVD). Our short time-gated measurements allow us to characterize in-situ the amplitude and time delay of the temporally-separated reflections of the higher-order modes (see Supplementary Fig. S5 and Supplementary Section S4). We thus verify in-situ that at the chosen time delay the measured holograms are the result of only the fundamental mode contribution. The relative time delays of the higher-order modes can be varied by changing the fiber length or core design.

The image contrast of FiDHo is a coherent reflection contrast, and has the same origins and limitations as in OCT. The image contrast and contrast-to-noise limits are analyzed in depth for OCT in[40,41]. In short, the image contrast-to-noise is dictated by the target reflectivity, index contrast, tissue scattering, laser coherence length, number of frames used for phase-shifting, and the camera pixel well-depth. In addition to these conventional limits, FiDHo imaging contrast also depends on the spurious reflections present in the system, inter-core crosstalk, and the applied digital Fourier filtering (see Methods).

Speckle noise is present in all results due to the coherent imaging nature of FiDHo, in a similar fashion to wide-field OCT. One reason is the nature of the reconstruction, where even a flat-phase target may introduce speckles: as the hologram signal from two neighboring cores is back-propagated, the non-delta response function of the system causes some spreading of the signal, and the overlap causes the formation of subjective speckle. In addition, multiple reflections and scattering present in biological tissues cause objective speckle formation.

To alleviate some of the speckle noise, similar approaches used in full-field OCT can be utilized. For example, incoherent compounding can be used, where the illumination and reference beam will be scanned over different MCF cores, creating multiple independent reconstructions of the same object. Coherent compounding of such results can improve the imaging resolution. An additional solution is incorporating computational denoising into the reconstruction algorithm e.g. using a compressed sensing approach[36].

FiDHo can also be employed with MCFs having disordered cores positions, as we demonstrate in supplementary Fig S4. The random

spatial sampling of such MCFs may allow improved reconstruction via compressed-sensing reconstruction algorithms[36].

A single-shot implementation of distal holography can be realized by off-axis holography, where the reference illumination is injected through a core off the center of the MCF. However, such an implementation would reduce the space-bandwidth product and thus the imaged FoV. A larger FoV and improved resolution may be obtained by extracting additional information from the intensity distribution of higher order modes of each core[42,43]. Similar to conventional MCF-based imaging techniques, improved resolution can be obtained by the use of a GRIN lens at the distal fiber end. The signal to noise and minimal detected reflectivity can be improved by using cameras with higher frame-rates and larger well-depth, as well as shorter coherence length sources[41]. The image reconstruction fidelity and resolution can be improved by employing compressed-sensing reconstruction algorithms that incorporate prior knowledge on the target object structure[36].

We have provided a proof-of-principle demonstration and in-depth investigation of a novel endoscopic imaging technique based on coherent elastic scattering contrast. As in OCT, such endogenous reflection contrast can be used for the diagnosis of various biomedical parameters, e.g. the detection of abnormal cells in the gastrointestinal tract[44]. Such a study requires the use of a shorter coherence-length source and potentially higher well-depth cameras and will be at the focus of future work.

## Methods
### Experimental design
Figure 1. depicts a simplified sketch of the experimental setup. The full experimental setup is presented in Fig. S1 and described below. To allow maximal flexibility in optical alignment and tuning of the various parameters, such as power splitting, the experimental setup used for our proof of principle experiments is based on a rather large number of bulk optical components. A simpler, more applicable design can be realized by using a single-mode fiber (SMF) based balanced interferometer design.

The illumination is provided by a diode laser at a wavelength of $\lambda = 640$ nm and coherence length of $l_c \approx 400$ μm (iBeam-smart-640s, Toptica, ∼ 1 mW at proximal facet). The illumination beam is split by a Mach-Zehnder Interferometer (MZI) using a polarizing beam splitter (PBS1, Thorlabs PBS251), and a non-polarizing 50-50 beam splitter (BS2, Thorlabs BS013). The powers ratio in the two arms is controlled by a half wave-plate (HWP1, Thorlabs WPH10M-633) placed before PBS1, and the polarizations are re-aligned by a second half-wave plate (HWP2, same as HWP1). The two arms need to be perfectly aligned, such that the two delayed beam replicas couple to the fundamental mode of a single MCF core. This is assured by using two separate 4-f telescopes (L1, L2. total of four $f = 100$ mm Thorlabs LA1509) placed in each of the MZI arms.

The two aligned beams exiting the MZI are coupled into a single core of the MCF using another telescope (L3, Thorlabs LB1901, $f = 75$ mm and L4, Thorlabs LA1979, $f = 200$ mm) and a long working-distance objective (OBJ1, Mitutoyo 378-803-3, M Plan Apo 10x, 0.28 NA). A non-polarizing 50-50 beam-splitter (BS3, Thorlabs BP150) is placed between OBJ1 and the MCF proximal facet. The MCF proximal facet is imaged through BS3 onto a camera (sCMOS, Andor Zyla 4.2Plus) by another long working distance objective (OBJ2, same as OBJ1), a lens (L5, Thorlabs LA1979, $f = 200$ mm) and a telescope with tunable magnification (ZOOM, Navitar 12X Zoom Lens System).

In order to minimize the proximal facet reflections in the camera image, two orthogonal linear polarizers (LP1, LP2: Thorlabs LPVISE100) are placed on the two ports of BS3 to effectively perform cross-polarized detection.

The reference arm length in the MZI is controlled by a two-mirror delay-line using a fine piezo-motor translator (Thorlabs PIA25 actuator and TIM101 controller) mounted on a larger translator (Thorlabs

Z825B actuator and KDC101 controller) for phase-shifting and time gating, respectively.

At the distal end, the small partially reflecting mirror surface is placed at a distance of $z_m$ = 2 mm from the fiber distal facet. The partial reflective mirror was fabricated by E-beam evaporation (EBPVD) of a 10 nm thick layer of Titanium on a 1 mm glass slide. An additional 1 mm-thick glass slide was used as a spacer, and an optical immersion oil with $n$ = 1.52 refractive index was placed between the glass spacer and the fiber and mirror to reduce unwanted reflections. The targets were placed at different distances behind the partially reflecting mirror, either by holding the samples in air or with immersion oil or water between the mirror and the samples. An additional distal camera (CMOS, Allied Vision Mako U-130B) is used for acquiring the ground-truth images of the sample using a microscope objective (OBJ3, Olympus UPlanFL n, 10 × 0.3 NA) and a lens (L6, Thorlabs LB1676, $f$ = 100 mm).

### Suppression of spurious reflections

Since the system is designed to image weakly reflecting objects, it is very sensitive to unwanted and spurious reflections from the MCF proximal facet. The cross-polarized detection effectively reduces most of the proximal reflective background. Thanks to random birefringence of the MCF cores[45], the reflected signal from the object is randomly polarized after propagation through the MCF, and thus the cross-polarized detection only halves the reflected object power on average. Spurious reflections at the MCF distal facet affect the single-excited core, which is straightforward to spatially filter digitally.

Two additional sources of undesired reflections that are naturally present but do not appear in the simple form of Eq. (1) are the reflection of the reference beam from the object, and the reflection of the illumination beam from the distal mirror. The interference terms arising from these two unwanted reflections are effectively suppressed by low-coherence time gating, as we explain below. In general, the intensity pattern on the fiber distal facet is the result of interference of four reflected fields: two desired fields and two undesired fields. The two desired reflected fields are the reflection of the reference beam from the distal mirror, $E_{R,m}$, and the reflection of the object illumination beam from the object, $E_{I,o}$. The two undesired reflected waves are the reflection of the reference beam from the object, $E_{R,o}$, and the reflection of the object illumination beam from the distal mirror, $E_{I,m}$. Thus, the intensity pattern at the fiber distal (and proximal) facet is:

$$I = |E_{R,o}e^{i\varphi_n} + E_{R,m}e^{i\varphi_n} + E_{I,o} + E_{I,m}|^2 \qquad (3)$$

where $\varphi_n$ is the phase-shifting phase.

The phase-shifting reconstructed hologram will be composed of 4 interference terms:

$$\underbrace{E_{I,o}E_{R,m}^*}_{A} + \underbrace{E_{I,m}E_{R,m}^*}_{B} + \underbrace{E_{I,o}E_{R,o}^*}_{C} + \underbrace{E_{I,m}E_{R,o}^*}_{D} \qquad (4)$$

The first term (A) is the desired signal, composed of the interference of the object illumination beam reflected from the object with the reference beam reflected from the mirror.

The three undesired terms (B-D) are as follows: The second term (B) arises from interference of the two reflections from the mirror and produces a large coherent background. The third term (C) results from two reflections from the object itself, and the last term (D) is the conjugate of (A) and will result in a defocused coherent background in the reconstruction. Without coherence-gating it would be very challenging to impossible to filter out the undesired interference terms. However, the four interferences occur at four distinct time delays between the reference arm and the object illumination arm, $\tau_A = \frac{2z_o}{c}$, $\tau_B = 0$, $\tau_C = 0$, $\tau_D = -\frac{2z_o}{c}$.

Using a source with a coherence length of $l_c \ll 2z_0$ allows to effectively suppress the unwanted interference terms by setting the time-delay between the two arms to $\tau = \tau_A$. A shorter coherence-length source would be beneficial to both improve the axial-sectioning resolution and reduce the minimal working distance. The shortest coherence length that can be used in FiDHo that provides interference in all fiber cores is determined by the fiber diameter and distal mirror distance. For the experimental parameters of the presented system the coherence length can be as short as ~20 μm.

### Digital filtering of MCF pixelation

Due to the low fill-factor of the MCF cores, the holographically-recorded field at the fiber facet plane is pixelated, and is effectively under-sampled by the MCF cores (Fig. 1B, inset). This under-sampling manifests itself as aliasing in the angular frequency domain, which results in "ghosts" replicas in the back-propagated reconstructed images (Fig. S2C). These artefacts can be suppressed by digitally filtering the reconstructed fields, as explained in detail in Supplementary Section S3. Briefly, we apply a low-pass filter on the sampled fields before back-propagation (Fig.S2). This effectively performs a Fourier-interpolation of the holographically measured fields between the cores at the fiber facet (Fig. S2D). We set the cutoff frequency of the low-pass filter to filter the aliased spatial frequencies above $k_{cutoff} \approx \pi/p$, where $p$ is the core-to-core pitch. The Fourier interpolation also effectively limits the detection $NA$, which dictates the reconstructed fields resolution. Similar to other works employing MCFs for imaging, since the $NA$ of the fundamental mode in each core is larger than $k_{cutoff}/2\pi$, some aliasing artefacts will always be present if ordered-cores MCFs are used, and may be avoided by using MCFs with aperiodic arrangement of cores[46]. In FidHO, the aliasing artefacts are naturally suppressed by the minimal value of the object distance from the fiber facet, set by the reference mirror distance. The minimal object distance and the fact that the illumination is provided by the fundamental mode of a single core leads to an inherent limit on the maximal angle (NA) of the reflected fields at the fiber facet (see Supp. Fig. S9). For an in depth discussion of digital filtering, please refer to Supplementary Section S3.

The presented results were obtained using an MCF with identical cores arranged on an ordered grid with very low crosstalk (Schott 153385). We obtained similar results using an MCF with inhomogeneous cores arranged on an imperfectly ordered grid (Fujikura FIGH-06-300S) (Fig. S4). In this case of an imperfectly ordered grid, the spatial cutoff frequency of the Fourier interpolation filter was set according to the average pixel pitch. A discussion on the effects of the various fiber parameters is presented in Supplementary section S2.

### Statistics and reproducibility

The experiments shown throughout this article were conducted at least 3 times, with similar results. Specifically, the results shown in Fig. 4, were conducted more than 3 times, on multiple glass slides and at different times. The results were similar, and a cell that was isolated enough from its neighbors was selected to be shown. The cells used in Fig. 4 were collected after written consent from a single contributor, using a cotton swab.

All numerical analysis and plots were created in MATLAB 2018. To reproduce results similar to those shown in Fig. 2, an example MATLAB code is provided along with a sample data set (see Supplementary Data 1).

## Data availability

All data needed to evaluate the conclusions in the paper are present in the paper and/or the Supplementary Materials. All experimental data are saved on a local server at the Hebrew University and available upon request from the authors.

## Code availability

Supplementary Data 1 contains an example MATLAB script and an experimental dataset, that can be used to recreate FiDHO results.

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

## Acknowledgements

This work received funding from the European Research Council (ERC) Horizon 2020 research and innovation program (grants no. 677909, 101002406, OK), Azrieli foundation, Israel Science Foundation (1361/18, OK), Israeli Ministry of Science and Technology (Grant 712845, OK). We thank the Hebrew University Center for Nanoscience and Nano-technology for distal mirror fabrication, and Tali Brooks for proofreading the manuscript.

## Author contributions

N.B. and O.K. conceived the idea and performed analytical analysis, N.B. performed numerical simulations, experimental measurements, and data analysis. N.B. and O.K. wrote the manuscript.

## Competing interests

The authors declare no competing interests.
