## [Peer Review File · Nature Communications]

REVIEWER COMMENTS

Reviewer #1 (Remarks to the Author):

The submitted paper titled “Real-time holographic lensless micro-endoscopy through flexible fibers via Fiber Bundle Distal Holography” offers a clever approach with great perspectives for minimally-invasive lensless fiber endoscopes with high robustness towards biomedical applications. The authors address the hurdle of the time-varying optical transmission function in coherent fiber bundles/ multicore fibers, MCF. Phase-shifted holograms of the specimen were captured by using a tiny partially reflective mirror and double shot, short coherence illumination. The written form of the work is at a high level, the physical background are described in detail. The figures are instructive. The paper is recommended for publication after successful revision.

Major comments:

-As stated in Eq. (1) the shape of the reference beam needs to be taken into account for object reconstruction. It is known, that MCF can support higher modes next to the base mode and that modal weights are bending dependent. How does this effect the imaging quality?

-In the discussion (line 193) and the supplementary (line 117) low inter-core crosstalk is mentioned as a desired fiber parameter. How and why does crosstalk influence the imaging quality?

-The authors claim real time capability of videos and give a framerate of 50 FPS. However, phase-shifting with multiple images (at least 3) is required. What is limiting the frame rate currently? Is single-shot off-axis holography possible?

-More discussion is required regarding the imaging capabilities: The authors should discuss axial resolution as well as lateral resolution, only an upper bound of 300 μm is given in Fig 6. Why is the lateral resolution limited to around 3 μm , when higher resolutions around or even below 1 μm were shown for scanning endoscopes based on MCF. Experimental data of FOV and σ_{PSF} for distances exceeding 500 μm could be given. What does image contrast depend on?

-Imaging of thick chicken breast tissue was demonstrated. However, only elastic scattered light can be used. Diagnostic cases with biomarkers should be presented where, for example, diseases can be identified from elastically scattered light.

-Crucial for applications is the image quality. There should be higher diffraction orders of the multi-core fiber, MCF occurring, which would limit the field of view. What is the value of the space-bandwidth-product? Approaches with aperiodic core structures were suggested, e.g. Extended field-of-view in a lensless endoscope using an aperiodic multicore fiber, Optics Letters Vol. 41, Issue 15, pp. 3531-3534 (2016).

Further comments:

-The cut-off frequency of the low-pass filter of $k_{cutoff} = \pi/p$ should be motivated.

-Fig. 1 outlines as parameters of the laser source, 640 nm wavelength, 200 μm coherence length. At page 6, methods, 400 μm coherence length is mentioned. Which value is correct?

-Fig 4 shows a high speckle background. Is this to be expected at biological tissue? Can this be resolved? What does the speckle pattern depend on? This should be addressed.

-Methods, "At the distal end, the small partially reflecting mirror surface is placed at a distance of $z_m=2\text{ mm}$ from the fiber distal facet. The partial reflective mirror was fabricated by E-beam evaporation (EBPVD) of a 10nm -thick layer of Titanium on a 1 mm glass slide". Regarding the ideal placement and dimensioning of the distal reflector several questions came up. As far as I understand, the realized setup employs macroscopic cover slides as spacer and reflector. In a real-world application this would need to be miniaturized to not enlarge the endoscope diameter. How would this be realized and how would this affect the transfer function of the distal optics, for instance due to reflexes at the lateral surface? 2 mm distance of the partial reflecting mirror and 270 μm diameter of the fiber results in very small triangulation angles, therefore low axial resolution and signal-to-noise-ratio (SNR). What is the achieved axial resolution and the SNR?

-How to discriminate signals of scattering in blood and water layers from the tissue surface, if the coherence gating cannot be used (scattering layers smaller than the coherence length)?

-The described optical setup on the proximal side is highly complex considering the primary task is to realize two spatially coinciding and temporally separated optical beams, only. The authors give sound reason regarding the additional components such as balancing of intensity and suppression of reflexes. The authors should discuss the influence of imbalanced intensity and reflexes onto the image quality in more detail.

Reviewer #2 (Remarks to the Author):

The paper from Badt and Katz reports on an interesting scheme to achieve amplitude and phase resolved refractive index imaging through a fiber optic bundle.

Using low coherence full-field phase shifting interferometry is not new and its implementation has been performed through optical fiber. What is novel here is its implementation through a fiber bundle to perform wide field imaging. Although the interest of such technique might remain limited (due essentially to the poor image quality) I think the idea is novel enough such as its may be published. The paper is clear and well written and the supplementary data are very helpful to discuss the various important parameters and limitations of the proposed technique.

There are few points the authors could address in a revised version of the manuscript to improve its clarity.

1- Reconstruction process: It is said that E_{illum} is assumed to be a Gaussian beam, however the fiber bundle used by the authors are known to have bi-modal structures in some cores. Did the authors observed cases where the reconstruction was odd and did they post-select results coming from reference fields emerging from a single mode fiber core?

2- The digital filtering of the MCF pixilation and its effect on imaging is not very clear to me. (1) Could the author rephrase this key sentence "by Fourier-interpolating the holographically measured fields, effectively interpolating the fields between the cores at the fiber facet". Also what are the effect on the imaging to filter out the replica, I guess there is a loss of brightness that leads to dimmer contrast? In general, it would be valuable to discuss more deeply the effects of this digital filtering.

3- In Fig 4 the cell can be viewed only in the phase contrast, this is quite limiting for the applications and this is related to the minimal reflectivity that the sample must produce to make the technique viable. Could the authors discuss what would be possible to achieve to improve the minimal reflectivity that the FiDHo technique can achieve.

4- Fig 6 on the chicken breast is not informative at all, rather it put the technique down for useful imaging. Could the author image a more interesting 3D sample (a mm insect, a pollen grain...)

Minor: the acronym FiDHo is not defined in the text (only in the title)

Reviewer #3 (Remarks to the Author):

The work by Noam Badt and Ori Katz presents an unpixellated video-rate micro-endoscopy imaging technique that does not require bulky optics at the distal end, is insensitive to bending distortions, capable of axial sectioning and with diffraction limited lateral resolution over an extended distance behind the distal fibre facet/partially reflective mirror. I find the work highly original, with an intriguing set of imaging characteristics not matched by any other available micro-endoscopy approach. Furthermore, the presented method has, in my opinion, a strong potential to be quickly adopted in a clinical setting due to its relative simplicity.

The paper is very well written, the principles of the method are clearly explained and the claims are supported by experimental evidence. I would also like to thank the authors for providing useful information for someone who would like to reproduce the work, including thorough discussion of unwanted back-reflections in the system, optimal reflectivity of the distal mirror and the optical setup design and sample considerations.

Based on the above I recommend the manuscript to be published in Nature Comm. However, I have several comments that I think would further improve the quality of the manuscript if addressed:

1. The claim of 3D imaging, and even “3D diffraction-limited” (page 5, line 183) is not the best choice of words given the current data. The system has axial sectioning capability, with experimental evidence hinting at 300um axial sectioning resolution. That is not 3D diffraction limited. I also think a more thorough discussion of the axial sectioning resolution achievable with the system is in order. At the moment, the reader has to look up coherence length of the laser in the text and make estimates of axial sectioning capability from such scattered information. I would like to see a transparent discussion of axial sectioning resolution from the authors.
2. Related to point 1. There is discrepancy between coherence length of the laser $l_c = 400 \text{ um}$ (page 6, line 216) in the text and 200um in Fig. 1, which makes the axial sectioning capability even harder to extract from the paper.
3. How miniature is the partially reflective mirror? Based on information in the Methods, it looks like the miniature mirror is made of two 1mm thick glass-slides, with immersion oil between the facet and the glass slides. I think that since the manuscript repeatedly claims how other methods suffer from bulky optics, more care should have been given to this issue. From my point of view, I think the partial mirror lateral extent can indeed have the diameter of the MCF – the method would work with that, and I do not mind the 2mm extra distance needed between MCF and mirror plus additional 2mm for glass slides. That can all become part of MCF at some point. However, all the current experiments were done with bulky glass slides if I am not mistaken. I understand that this is just a demonstration and making a miniature mirror is a purely engineering problem. However, it should be clear that the experiment was done with bulky glass slides and a hint of a pathway how to make a small add-on on MCF should be part of this paper.
4. Page 3, Line 114, equation (2). It would be beneficial to mention that the propagator can deal with different materials along the way, immersion oil/air etc.
5. Page 4, Line 124, correct to “at the different axial plane”
6. Page 4, Line 144, can you explicitly write the z_{obj} so that the end-user can get a feeling for the maximum distance over which the resolution and FOV do not change?
7. I would move the Phase-contrast imaging paragraph at the very end of Result section.
8. Page 4, Line 167 correct to “collected through”
9. Is there a way to quantify how insensitive the imaging is to bending?
10. Fig 2A. I see LP11 like modes in the inset. Why is the light coupled as an LP11 mode? Can the presence of LP11 influence the reconstruction of the object field?
11. Fig.1A the units of laser are as subscripts

12. Page 5, line 192. You claim bend sensitivity is minimised by using single mode fibers, yet Fig 2A shows your fibres are multimode and couple LP11 in some cases. Can you please discuss?
13. Page 5, line 196, correct "reflectivity" twice in the same line
14. Fig 3J, why is the minimum distance of the object from mirror 60um or so? Is it the limitation of the system or of the measurement method? Also, it looks like the method is more noisy for smaller z_0 . At least that is my impression from Fig S3 but I might be wrong. I don't see any reason why that would happen. Can you please comment?
15. I find Fig 6B lacking information. I know this might be a bit too much to ask but I think a much nicer demonstration would be to make a sample with defined etched steps and then do the stitching on this. For me, that would be a much better proof of depth sensing. I actually think Fig A is enough. If B is there just to push some biology in, I would say remove it, because it has little meaning in its current form. I also think that axial sectioning resolution can be discussed at this point, including limitations and that choosing a laser with shorter coherence length might improve axial sectioning.
16. Supplementary: Page 3, Line 70 correct "reflectivity"
17. Supplementary: Page 4, Line 105 correct "10 nm". nm as subscript
18. Fig S1: There is some spurious s under figure caption
19. Fig S3C: the $z_0 = 90\mu\text{m}$ really looks noisier. Any reason why?
20. Fig S4: It was not clear to me why this was included. Maybe a short discussion why imaging using these fibers is important would suffice.

Martin Ploschner

Reply to reviewers,

“Real-time holographic lensless micro-endoscopy
through flexible fibers via Fiber Bundle Distal Holography (FiDHo)”

In our reply below we cite the referees' comments in *green*, our replies appear in *blue*, and the corresponding revisions made to the manuscript appear in standard text. The revisions made to the manuscript appear in *blue* in the manuscript text.

Reviewer #1

The submitted paper titled “Real-time holographic lensless micro-endoscopy through flexible fibers via Fiber Bundle Distal Holography” offers a clever approach with great perspectives for minimally-invasive lensless fiber endoscopes with high robustness towards biomedical applications. The authors address the hurdle of the time-varying optical transmission function in coherent fiber bundles/ multicore fibers, MCF. Phase-shifted holograms of the specimen were captured by using a tiny partially reflective mirror and double shot, short coherence illumination. The written form of the work is at a high level, the physical background are described in detail. The figures are instructive. The paper is recommended for publication after successful revision.

We thank the referee for his positive view of our work.

Major comments:

As stated in Eq. (1) the shape of the reference beam needs to be taken into account for object reconstruction. It is known, that MCF can support higher modes next to the base mode and that modal weights are bending dependent. How does this effect the imaging quality?

We thank the referee for raising this point, as indeed the MCF cores support several modes. In our experiments, in order to assure that the fundamental mode is used as the reference beam, we have utilized the short (~1 picosecond) coherence length to selectively time-gate and measure only the interference of the object field with the fundamental mode, which is the first arriving mode. For the fiber used in our experiments the fundamental mode arrival time is ~3ps prior to the second mode, as we determined in-situ from the time-delay scan holographic measurements (see attached Figure below).

The in-situ time-resolved holographic measurements of the reference mirror reflection allow us to quantify the relative weights and delays of the higher modes (see attached new revised figure). As noted by the referee, this is crucially important in order to assure that the fundamental mode is indeed used as the reference beam, as well as to assure that the contribution of the interference of the higher modes reference and object fields are sufficiently small. In our experiments, as can be seen in the new revised figure below, ~2% of the measured time-gated energy originates from the higher order mode, when a time delay of ~5ps is selected. This residual energy will be manifested by a $\sim(0.02)^2$ contribution to the measured holograms when the time-gate for the first mode is selected, since the 2% power of the second order mode reflection from the object will interfere with the ~2% power of the second order mode reflection from the reference mirror.

To address this important point in our revised manuscript we have added a new figure of the experimental time-resolved measurements of the first and second mode to the Supplementary section (see below), and the following discussion added to the revised manuscript main text:

Our approach assumes that both the object illumination and the reference beam illumination originate from the fundamental mode of the fiber bundle core. While each core supports also

higher-order modes, their measured arrival times are temporally separated by delays that are introduced at the illumination path by the fiber modal group velocity dispersion (GVD). Our short time-gated measurements allow us to characterize in-situ the amplitude and time delay of the temporally-separated reflections of the higher-order modes (see Supplementary Fig. S5 and Supplementary Section S4). We thus verify in-situ that at the chosen time delay the measured holograms are the result of only the fundamental mode contribution. The relative time delays of the higher-order modes can be varied by changing the fiber length or core design.

To complement the experimental time-resolved measurements of the different modes, we have numerically investigated the effects of residual higher modes interference on the reconstructed images. Both the experimental and numerical results appear in the new Supplementary Section, which reads:

Section S4: Analysis of the effects of higher-order modes in each core

As each core of the commercial MCFs used in our work supports several transverse modes, in our experiments care was taken to excite only the fundamental mode by the illumination and reference beams. However, such excitation is not perfect, and in addition fiber bending and manufacturing imperfections may couple the beams injected in the fundamental mode to higher modes. This requires additional care in the measurements and analysis to the possible contribution of higher order modes of each core to the image formation. The most important and very effective experimental measure to minimize the contribution of the higher order modes is our use of the short coherence length source to time-gate the interference with the fundamental mode, which temporally arrives before the higher order modes. This was proved to be sufficient to reduce the contribution of higher order modes to a negligible measure. To illustrate this, below, we experimentally characterize the higher order modes in our experiments, and numerically analyze the potential effects of higher order modes on the reconstructed images.

Experimental measurements

To experimentally characterize the temporal delay and relative amplitude of the second transverse fiber mode, we performed a measurement of the total energy of the recorded hologram as the relative delays between the two interferometer arms (the ‘object illumination’ arm and the ‘reference’ arm) is scanned, for a target object placed at a distance of $200\mu\text{m}$ from the distal mirror. The results of this scan are displayed in Fig S5.A.

As expected, the measured scan displays three dominant peaks (i,ii,iii): the most dominant peak (i) appears at zero relative delay, and is dominated by the interference of the strong reflection from the distal mirror in one arm with the same reflection in the second arm. In addition, it contains the interference of the illumination field reflected from the object with the reference field reflected from the object. As we explain below, when a second mode is present, additional mirror-mirror and object-object interferences are contributing to the hologram due to the second mode (see Fig.S5E and explanation below). The second time-gated interference peak (ii) is the desired interference between the fundamental mode illuminating the object and the fundamental mode reference beam reflected from the mirror, used for imaging in our experiments. This peak appears when the delay is set to the object distance: $\tau = 2z_o/(c/n)$. The third time-gated interference peak (iii) is the result of the reflections from the second mode in the illumination arm with the fundamental mode in the reference arm (see temporal diagram in Fig.S5E).

In accordance, the spatial shape of the holograms measured at these three delays reveal the nature of the different fields that take part in each interference: for (i) the hologram reveals the distal mirror flat-phase reflection (Fig.S5B); for (ii) the hologram reveal the object reflected

field on the facet (Fig.S5C), which is used for image reconstruction; for (iii) the hologram reveals the second-order mode illumination of the distal mirror, as measured by the fundamental mode (Fig.S5D). Most importantly, this experiment displays how the short time-gating allows not only to filter the strong mirror reflection, but also to effectively filter the unwanted contributions of the second (and higher) modes.

To clarify the different expected contributions to these three interference peaks (i-iii), we have plotted in Fig.S5E a sketch of the reflections from the distal mirror and object for the first two modes in each arm, for the three different delays (i-iii).

As in Supplementary Section S2, we denote the fields of the reference arm by subscript (R), the fields from the illumination arm by subscript (I), the reflection from the distal mirror or object are denoted by subscripts (m,o) respectively. The first and second mode are denoted by superscript (1) and (2). The fields generated by the reference arm are thus: $E_{R,m} = E_{R,m}^{(1)} + E_{R,m}^{(2)} + E_{R,o}^{(1)} + E_{R,o}^{(2)}$, and the fields generated by the illumination arm are thus: $E_{I,m} = E_{I,m}^{(1)} + E_{I,m}^{(2)} + E_{I,o}^{(1)} + E_{I,o}^{(2)}$. As result of the short time-gate, the fields that contribute to each of the interference peaks, are only those that temporally overlap, as marked by the dashed vertical lines in Fig.S5E. At the desired delay for imaging (ii), the measured hologram is the result of the interference of the fundamental-mode reflection from the distal mirror of the reference beam with the reflection of the fundamental mode of the illumination beam from the object, as desired: $E_{I,o}^{(1)} E_{R,m}^{(1)}$. The relatively weak interference of the second mode illuminating the mirror and object ($E_{I,o}^{(2)} E_{R,m}^{(2)}$) is coherently added to the strong hologram of the first mode. The interference due to the second mode is very small in magnitude since its amplitude is squared in the recorded hologram (both reference and illumination fields are excited weakly in the higher order mode). From the measured experimental trace (Fig.S5A) we estimate that the energy in the second order mode is approximately 2% of the fundamental mode energy.

Fig S5: Effects of higher-order fiber-modes on time-gated holograms. (A) Total phase-shifted hologram energy vs delay introduced between the reference- and illumination-arms for a USAF-target placed at $z_{obj} = 205\mu\text{m}$. Three dominant peaks (i,ii,iii) are observed when the illumination delay matches either the zero relative delay (i), in which the reflected illumination (‘object field’) is dominated by the reflection from the distal mirror, the reconstructed hologram at this delay is a flat-phase and amplitude one (B). The second peak (ii) occurs when the delay matches the object distance $\frac{z_o n}{c}$, in which the object field is dominated by the first mode reflected from the target, and the reconstructed hologram is the object diffraction (C). The third peak (iii) occurs when the delay is equal to the difference between the second and first fiber mode arrival times, in which the distal object field is dominated by the reflection of the second mode from the distal mirror, and the reconstructed hologram is the diffraction of the second mode (D). Sketch (E) depicts the four delayed beams at the distal tip for each of the two arms, considering two fiber modes. Triangles and rectangles represent the mirror and object reflection respectively, solid and dashed lines represent the first and second mode respectively and red and yellow represent the reference and illumination arm respectively. The three peaks (i,ii,iii) are displayed as three different illumination delays. From the experimentally measured relative energies, we retrieve the approximated values for the reflected intensity from the object: $R_o = 0.6\%$, and the second mode relative intensity: $M_2 = 2\%$

Numerical investigation of higher-order mode impact on reconstructed images

In the previous section, we have shown that the experimentally measured hologram at the optimal time-delay is the sum of two interference terms: $E_{I,o}^{(1)} E_{R,m}^{(1)} + E_{I,o}^{(2)} E_{R,m}^{(2)}$. The first is the desired hologram generated by the fundamental mode, and the second is a significantly weaker undesired hologram that is generated by the second mode. Experimentally, the weak undesired hologram contribution is difficult to measure and separate from other noise and background contributions. Thus, in order to complement the experimental investigation and characterization of the higher-order modes, we have performed a numerical simulation where we study the impact of the contribution of the second mode on the reconstructed image.

The results of this numerical investigation are displayed in Supplementary Figure S6. The results are obtained from a simulation based on digital angular spectrum propagation, which is performed separately for each mode of the MCF core. The holographic recorded field is simulated in several steps: (i) the illumination field from a selected single mode ($j = 1, 2, \dots$) from a single core at the MCF distal facet is propagated to the object plane; (ii) the field illuminating the object is multiplied by the 2D object reflection function; (iii) the reflected field is propagated back to the MCF distal facet to produce the object field from the illumination arm at the distal facet for each mode, j : $E_{I,o}^{(j)}(x, y)$; (iv) the same steps are performed for all the fiber modes studied when replacing the object reflection function and distance by the distal mirror reflection, to simulate the reference field at the distal facet for each mode: $E_{R,m}^{(j)}(x, y)$. The simulated holographically measured fields are then calculated for each mode by:

$$E_{holo}^{(j)}(x, y) = \left(E_{R,m}^{(j)}(x, y) \right)^* E_{I,o}^{(j)}(x, y)$$

Following (Eq.1), the reconstructed object field image is produced by normalizing the holographic recorded field by the expected diffraction of the reference mode $\left(E_{R,m}^{(1)}(x, y) \right)^*$, followed by back-propagation:

$$O^{(j)}(x, y, z_{prop}) \propto \mathcal{P}_{-z_{prop}} \left(\frac{\left(E_{R,m}^{(j)}(x, y) \right)^* E_{I,o}^{(j)}(x, y)}{\left(E_{R,m}^{(1)}(x, y) \right)^*} \right)$$

For the first mode contribution ($j=1$), the normalization is correct, and the reconstructed image is the same as in the ideal single-mode case. For other modes, the normalization does not correctly compensate for the contribution of the reference mode, and the holographic recorded

field is multiplied by an erroneous factor of: $\frac{\left(E_{R,m}^{(j)}(x, y) \right)^*}{\left(E_{R,m}^{(1)}(x, y) \right)^*}$ (in addition to a different illumination of the object), which results in imaging artefacts in the reconstruction fields that are a coherent sum of the different modes contribution: $O_{reconst}(x, y) = \sum_j O^{(j)}(x, y, z_{prop})$.

Fig. S6 displays simulated reconstructed object fields for a USAF target for different relative amplitudes of second-order transverse mode excitation. As noted in the previous section, we estimate the energy of the second order mode in our experiments to be $\sim 2\%$ of the fundamental mode energy. Importantly, we note that the contributions of the higher-order modes is not a noise-like or background contribution, but a coherent term that contains imaging information. As such, it may be used for improved imaging by digitally taking into account the distribution of the reference field in the higher modes: $E_{R,m}^{(j)}(x, y)$, and the difference in object illumination. This may be useful to improve the imaging NA and potentially the FoV, but will require a more

involved reconstruction scheme, to simultaneously determine the object pattern and the modal excitation complex amplitudes.

Fig S6: Simulated effect of higher-order fiber modes on object reconstruction. Numerical investigation of the reconstructed object field when the relative delay between the illumination- and reference arms matches the object distance (i.e. case (ii) of Fig.S5). (A) Assuming only the fundamental mode LP_{01} is excited in both reference and illumination arms. (B) Assuming 2% of the illumination energy excites the second (LP_{11}) fiber mode in both arms, as in our experiments (see Fig.S5). (C) Displays the simulated contribution of the second mode (LP_{11}), to the hologram. (D,E,F) are zoom-in on the dashed boxes in (A,B,C).

1. -In the discussion (line 193) and the supplementary (line 117) low inter-core crosstalk is mentioned as a desired fiber parameter. How and why does crosstalk influence the imaging quality?

We thank the referee for raising the need for further discussion of this subject. In order to address this point we added the following paragraph to the supplementary section S2, under the subtitle “Fiber parameters”

Similar to fiber-bundle based imaging schemes, a lower inter-core crosstalk is important to the FiDHo approach. Since in FiDHo the fiber-bundle is used both in the illumination step and the collection step, in both reference and object arms, the influence of the inter-core crosstalk can be divided into several effects:

1. In the illumination path, a high crosstalk will result in leakage of the light to adjacent cores, thus creating a complex interference pattern in the reference and object illumination beams. The resulting effect is twofold:
 - a. The reference field is assumed to be the diffraction of the fundamental mode of a single core. Intercore crosstalk will result in an unknown reference field that is also varying and very sensitive to fiber bending.

- b. The object illumination would have dark areas due to destructive interference. In addition the object will be illuminated by an unknown spatially varying, and bend-sensitive phase, which would not allow true phase reconstruction.
2. In the collection path, similar to conventional fiber-bundle based imaging approaches, a large intercore crosstalk will cause the signal to leak to adjacent cores, reducing the resolution of the measured hologram image, and creating coherent interference between the different cores, that would be bend sensitive.

2. 3A) The authors claim real time capability of videos and give a framerate of 50 FPS. However, phase-shifting with multiple images (at least 3) is required. What is limiting the frame rate currently?

We thank the referee for this question. The limiting factor in our experiments was the sCMOS camera frame-rate. To address this point in the revised manuscript we have added the following discussion to the “Results” section, under the current subsection: “Video-rate and sensitivity to bending”:

Each FiDHo reconstructed image requires at least 3 phase-shifting camera frames. In the Supplementary movies, each 3 consecutive camera frames are used to reconstruct one image. Therefore, while the images are reconstructed at the camera frame-rate of 50 frames-per-second, two out of three camera frames are shared between consecutive reconstructions, i.e., there exist 16.6 unique reconstructed frames per second. In our experiments the imaging speed was limited by the sCMOS camera frame-rate. Improved imaging speeds can be obtained by using cameras with higher frame-rates and by replacing the piezo-stage by a faster phase-shifting mechanism, e.g. an acousto-optic modulator.

3B) Is single-shot off-axis holography possible?

Yes indeed. In fact, in our original vision of the technique we envisaged a single-shot widefield holographic endoscope, where the reference beam “guide-star” is injected from a core at the edges of the MCF, while the object illumination is performed from the central core. However, while off-axis holography is attractive for improving imaging speed, it has several fundamental drawbacks that are in particularly problematic for implementation through an MCF:

1. First, in choosing off-axis holography one is reducing the space-bandwidth product (number of imaging pixels) for reducing acquisition time. While this may be a natural choice for high pixel count cameras, this is not the case in commercial MCFs, which have a very small number of cores (“image pixels”) as compared to a modern camera pixel count. While this is possible, this would reduce the field of view to values that are not extremely useful for most applications.
2. Another point where MCFs differ from modern cameras is their relatively low fill-factor, which is reflected in a rather large core-to-core spacing in comparison to the fundamental mode size. This results in rather sparse spatial sampling of the holographically recorded fields, and limits the maximum angle of the off-axis reference beam due to the requirement of Nyquist sampling. The small relative angle of the off-axis reference beam, will limit the imaged object size (field of view) as the object must be sufficiently small (when placed in the far-field) to avoid overlapping with the twin-image, fundamentally present in off-axis holography.

At short relative working distances it would be challenging to provide a large, homogeneous reference intensity in all fiber cores with the illumination originating from one of the off-axis cores.

After careful theoretical and experimental consideration of possible single-shot implementation, we have concluded that for the planned experiments, the above disadvantages of off-axis holography make it a less desired solution for practical applications. Nonetheless it is definitely possible to realize such a measurement scheme.

In order to address this point in the revised manuscript, we have added the following statement to the revised Discussion section:

“A single-shot implementation of distal holography can be realized by off-axis holography, where the reference illumination is injected through a core off the center of the MCF. However, such an implementation would reduce the space-bandwidth product and thus the imaged FoV”

4. 4A)- More discussion is required regarding the imaging capabilities: The authors should discuss axial resolution as well as lateral resolution, only an upper bound of 300 μm is given in Fig 6.

We thank the referee for noting the missing discussion on the axial resolution. To address it we have added the following discussion and analysis to the “Results” section of the main text, under the subtitle “Resolution and Field-of-View”, and have added a measurement of the coherence length of the laser used as a new Supplementary Fig. S7. The additional discussion reads:

The axial resolution of FidHO is determined by two factors: the axial sectioning due to coherence gating, and axial resolution due to the numerical aperture of the imaging system. First, similar to OCT, the coherence gating yields an axial sectioning with an axial resolution dictated by the coherence length of the laser used, l_c . In our experiments $l_c = 400\mu\text{m}$. However, a shorter coherence length source can be used to improve the axial sectioning resolution. An experimental measurement of the coherence-gating axial sectioning in our experiments is presented in Supp.Fig. S7.

As in other holographic imaging approaches, in addition to the coherence-gating axial resolution, the imaging axial resolution that originate from the NA of the recorded holograms is dependent on the imaged object: for a point-reflector, the axial resolution would be $\delta z \approx$

$\frac{\lambda}{NA^2} \approx 30\mu\text{m}$. For a flat-phased object of diameter d_{obj} , $\delta z \approx \frac{d_{obj}^2 + \left(\frac{\lambda}{NA}\right)^2}{\lambda}$. The axial and lateral resolutions can be improved by advanced computational reconstruction schemes [36] (see Discussion).

Fig S7 Measured Coherence function of the laser source. Total energy of phase-shifted hologram vs. the relative delay between the illumination and reference arms. The measurement was performed on the two beams directly at the output of the interferometer. The measured full-width at half max (FWHM) is $400\mu\text{m} \pm 50\mu\text{m}$ (red).

4B) .Why is the lateral resolution limited to around $3\mu\text{m}$, when higher resolutions around or even below $1\mu\text{m}$ were shown for scanning endoscopes based on MCF. Experimental data of FOV and σ_{PSF} for distances exceeding $500\mu\text{m}$ could be given.

The lateral resolution is limited to approximately $3\mu\text{m}$, which is the diffraction-limited resolution of the recorded holograms, dictated by the effective NA of the fundamental mode of the core, which is used for both illumination and detection. To address this point we have revised the main text discussion lateral resolution to read:

The resolution is diffraction-limited by the numerical aperture (NA) of the recorded holograms, which is dictated by the NA of the fundamental mode used for illumination and detection, and by the digital low-pass filtering of the holograms at $f_{\text{cutoff}} = 1/2p$ that is performed to filter artefacts caused by pixelation (sampling) of the ordered MCF cores, where p is the core-to-core pitch (see Methods and Supplementary section S3).

While indeed, resolution of down to $1\mu\text{m}$ are reported using MCFs, these are usually achieved by placing a GRIN lens at the distal fiber end, which provides the required magnification and NA. While not studied in this work, FiDHo can also be implemented using a GRIN lens for improved resolution. To note this point we have added the following sentence to the main text Discussion:

Similar to conventional MCF-based imaging techniques, improved resolution can be obtained by the use of a GRIN lens at the distal fiber end

4C) *What does image contrast depend on?*

We thank the referee for pointing out the need for further discussion on image contrast. To address this point, we have added a discussion paragraph to the main manuscript text. In short,

similar to OCT, and holographic imaging techniques, the imaging contrast in our approach is reflection (amplitude and phase) contrast. The origins and the fundamental limits of the imaging contrast-to-noise are equivalent to those in OCT, and include refraction index contrast, power ratio between the reference and illumination arm, and camera well-depth. These are analyzed in-depth for OCT in reference [41] (also [41] of the original manuscript), as well as in [“Full-field optical coherence microscopy”, E. Beaurepaire, A. C. Boccara, M. Lebec, L. Blanchot, and H. Saint-Jalmes, Optics Letters 23, 4, 244-246 (1998)].

To address this point in our revised manuscript, we have added the following paragraph to the Discussion section, which includes a reference to “Full-field optical coherence microscopy”, as reference [40]:

The image contrast of FiDHo, is a coherent reflection contrast, and has the same origins and limitations as in OCT. The image contrast and contrast-to-noise limits are analyzed in depth for OCT in [40,41]. In short, the image contrast-to-noise is dictated by the target reflectivity, index contrast, tissue scattering, laser coherence length, number of frames used for phase-shifting, and the camera pixel well-depth. In addition to these conventional limits, FiDHo imaging contrast also depends on the spurious reflections present in the system, inter-core crosstalk, and the applied digital Fourier filtering (see Methods).

An analysis of the optimal reference wave relative intensity and mirror reflectivity that would maximize the signal to noise of FiDHo measurements is analyzed in Supp.Section S2.

5. Imaging of thick chicken breast tissue was demonstrated. However, only elastic scattered light can be used. Diagnostic cases with biomarkers should be presented where, for example, diseases can be identified from elastically scattered light.

We thank the referee for raising this important question. Indeed, elastic scattering endoscopic OCT was shown to be useful in e.g. the detection of dysplasia in the gastrointestinal tract, even without the use of exogenous biomarkers [“Criteria for the diagnosis of dysplasia by endoscopic optical coherence tomography” Pfau et al. Gastrointestinal endoscopy 58.2 (2003): 196-202.]. However, performing a similar study using FiDHo requires the replacement of the laser source to a shorter coherence source, and acquiring the appropriate biological specimens and approvals. While this is definitely a long-term goal of this work, it is unfortunately beyond the scope of our current work, which is focused on a proof-of-principle demonstration and in-depth analysis of a novel imaging approach. To address this point we have added the following sentence to the Discussion of the revised manuscript:

We have provided a proof-of-principle demonstration and in-depth investigation of a novel endoscopic imaging technique based on coherent elastic scattering contrast. As in OCT, such endogenous reflection contrast can be used for diagnosis of various biomedical parameters, e.g. the detection of abnormal cells in the gastrointestinal tract [44]. Such a study requires the use of shorter coherence-length source and potentially higher well-depth cameras and will be at the focus of future work.

Where [44] refers to “Criteria for the diagnosis of dysplasia by endoscopic optical coherence tomography” Pfau et al. Gastrointestinal endoscopy 58.2 (2003): 196-202”.

6. Crucial for applications is the image quality. There should be higher diffraction orders of the multi-core fiber, MCF occurring, which would limit the field of view. What is the value of the space-bandwidth-product? Approaches with aperiodic core structures were suggested, e.g. Extended field-of-view in a lensless endoscope using an aperiodic multicore fiber, Optics Letters Vol. 41, Issue 15, pp. 3531-3534 (2016).

Indeed, the diffraction orders of the ordered MCF cores result in aliasing in the k-space domain, as is displayed in the original manuscript Supp.Fig.S2 (see below). While such aliasing result in replicas in k-space domain, which directly limits the FoV if the imaged objects are at the far-field of fiber facet, in our work the imaged objects are located at closer distances and the k-space aliasing results in artefacts that are not localized at the image plane (see Supplementary Fig.S2 below). In our original manuscript, we discussed these artefacts in detail in the Methods section, and in Supplementary Fig.S2. Specifically, the aliasing artefacts inherent to ordered MCF cores, are suppressed in our work due to the minimum working distance set by the reference mirror, and digital low-pass filtering performed on the raw measured holograms at the fiber facet. As noted by the referee, these artefacts can be avoided by the use of specially-engineered aperiodic cores arrangements, as in the work of Sivankutty et al.

To address the referee's questions, we have rewritten the main text paragraph and Methods section that discuss these points, have calculated the space-bandwidth product of our system from the experimentally measured FoV and resolution, and have added a reference to the work of Sivankutty et al., as reference [46] in the revised manuscript. The revised manuscript paragraphs read:

Revised Methods section:

Digital filtering of MCF pixelation

Due to the low fill-factor of the MCF cores, the holographically-recorded field at the fiber facet plane is pixelated, and is effectively under-sampled by the MCF cores (**Error! Reference source not found.** B, inset). This under-sampling manifests itself as aliasing in the angular frequency domain, which results in "ghosts" replicas in the back-propagated reconstructed images (Fig. S2 C). These artefacts can be suppressed by digitally filtering the reconstructed fields, as explained in detail in Supplementary Section S3. Briefly, we apply a low-pass filter on the sampled fields before back-propagation (Fig.S2). This effectively performs a Fourier-interpolation of the holographically measured fields between the cores at the fiber facet (Fig. S2 D). We set the cutoff frequency of the low-pass filter to filter the aliased spatial frequencies above $k_{cutoff} \approx \pi/p$, where p is the core-to-core pitch. The Fourier interpolation also effectively limits the detection NA, which dictates the reconstructed fields resolution. Similar to other works employing MCFs for imaging, since the NA of the fundamental mode in each core is larger than $k_{cutoff}/2\pi$, some aliasing artefacts will always be present if ordered-cores MCFs are used, and may be avoided by using MCFs with aperiodic arrangement of cores [46]. In FidHO, the aliasing artefacts are naturally suppressed by the minimal value of the object distance from the fiber facet, set by the reference mirror distance. The minimal object-distance and the fact that the illumination is provided by the fundamental mode of a single core leads to an inherent limit on the maximal angle (NA) of the reflected fields that at the fiber facet (see Supp.Fig. S9). For an in depth discussion of digital filtering, please refer to Supplementary Section S3.

The presented results were obtained using an MCF with identical cores arranged on an ordered grid with very low crosstalk (Schott 153385). We obtained similar results using an MCF with inhomogeneous cores arranged on an imperfectly ordered grid (Fujikura FIGH-06-300S) (Fig. S4). In this case of imperfectly ordered grid, the spatial cutoff frequency of the Fourier interpolation filter was set according to the average pixel pitch. A discussion on the effects of the various fiber parameters is presented in Supplementary section S4.

Revised main text:

...as expected from the theoretical analysis (see section S1), the resolution and FoV are independent of imaging distance for distances of $z_{obj} < D_{fiber}/2NA - z_m$, where D_{fiber} is the

MCF diameter. The resolution is diffraction-limited by the numerical aperture (NA) of the recorded holograms, which is dictated by the NA of the fundamental mode used for illumination and detection, and by the digital low-pass filtering of the holograms at $f_{cutoff} = 1/2p$ that is performed to filter artefacts caused by pixelation (sampling) of the ordered MCF cores, where p is the core-to-core pitch (see Methods and Supplementary section S3). The FoV is half the fiber diameter. For our experimental parameters ($z_m \approx 2\text{mm}$, $D_{fiber} \approx 600\mu\text{m}$), the measured resolution and FoV ($250\mu\text{m}$, full width at half max) indicate an effective NA of $NA_{eff} \sim 0.15$, and a space-bandwidth product of $\sim 5,300$. In line with the fiber parameters (Schott 153385) and the interpolation performed (see ‘‘Digital filtering MCF pixelation’’ in Methods section).

To better address this point and provide more detailed information on the filtering/interpolation process, we have revised Supplementary Fig.S2 and its caption, which now read:

Fig. S2. Fourier Interpolation on ordered MCF (A) The measured distal field on a Schott fiber used throughout the article, shows the individual cores with a constant inter-core pitch, p , which perform effective spatial sampling of the field at a spatial frequency of $1/p$. (B) In the spatial frequency domain, replicas appear as result of the spatial sampling, as expected from the convolution theorem. The dashed red circle marks the spatial frequency range that is within the Nyquist sampling criterion ($|f| < \frac{1}{2p}$, i.e. $|k| < \frac{\pi}{p}$). (C) Reconstruction of the object directly from the measured field of (A), displays ghost replicas, due to the frequency-space aliasing. (D, E, F) show the same data, after low-pass filtering the aliased high spatial frequencies outside the dashed circle in (B). Scale bars: A, C - $100\mu\text{m}$ B - $\frac{2\pi}{p}$

Further comments:

7. -The cut-off frequency of the low-pass filter of $k_{cutoff} = \pi/p$ should be motivated.

To address this point we have added a new Supplementary Section discussing in depth the digital filtering of the holograms prior to back-propagation, with an explanation of the choice of the cutoff frequency and additional details. The new Supplementary section reads:

Section S3: Digital filtering of MCF pixilation

The distal holography approach is based on holographic recording of the field at the distal facet. However, since the field hologram is not sensed directly but is effectively sampled the fiber cores, having a limited fill-factor due to the core-to-core pitch being larger than the mode diameter, sampling-induced artefacts (resulting from aliasing in spatial frequency space), appear in the reconstructed field. To achieve the high image quality of FiDHo, these sampling-induced artefacts need to be suppressed. Here, we obtained the required suppression of these artefacts by a combination of imaging and illumination geometry and natural filtering of the MCF cores, which attenuated high spatial frequencies, in addition to digital filtering in post-processing. In this section we provide a mathematical and numerical analysis of the sources for the sampling-induced artefacts, and of the effectiveness of the approaches we have implemented to suppress them.

We first briefly present and explain the experimentally observed artefacts, as displayed in Supplementary Fig.S2a-c. In our experiments with a commercial MCFs produced by Schott, cores are located on a hexagonal grid with a core-to-core pitch, $p = 8\mu\text{m}$ (Fig.2A). Thus, the incident field hologram is effectively sampled at a spatial sampling frequency, $f_s = 1/p$. As result of the convolution theorem, in the Fourier (angular frequency) domain, this spatial domain sampling results in generation of replicas of the object angular spectrum at shifts of $\Delta f = m f_s$, where $m = \pm 1, \pm 2, \dots$ (Fig.S2B). If the object plane is located at a distance, z , which is effectively at the far-field of the MCF, these aliased replicas will limit the FoV to $\sim \lambda z/p$ [46]. If the object is located at shorter distances from the fiber, the angular frequency artefacts will results in object replicas inside the imaged FoV (Fig.S3C).

To yield the following mathematical analysis more accessible and without loss of generality, we perform the analysis in one dimension (1D). Extension to two dimensions (2D) is straightforward.

Consider an electric field, $E_{\text{signal}}(x)$, being measured through a 1D array of waveguides spaced evenly with a pitch p , representing the MCF cores. Each waveguide can guide light with a complex amplitude that is proportional to the overlap-integral of the electric field that impinges upon it with a certain mode M_d , of typical mode-field-diameter d . Therefore, the measured field on the other side of the array will be:

$$E_{\text{meas}}(x) = \left((E_{\text{signal}}(x) * M_d(x)) \cdot \text{comb}_p(x) \right) * M_d(x)$$

Where $*$ symbols a convolution that realizes the overlap integral with the fiber mode, comb_p is the Dirac-comb composed of delta-functions spaced at intervals of p .

In the angular frequency domain, this sampled field is given by:

$$\tilde{E}_{\text{meas}}(f_x) = \left(\left(\tilde{E}_{\text{signal}}(f_x) \cdot \tilde{M}_{1/d}(f_x) \right) * \text{comb}_{1/p}(f_x) \right) \cdot \tilde{M}_{1/d}(f_x)$$

Where the convolution theorem was used. Assuming that M_d is a gaussian-like function of typical width d we can assume $\tilde{M}_{1/d}$ to be of typical width $\frac{1}{d}$. We note that in an MCF the core diameter is smaller than the core spacing: $d < p$.

Several conclusions can be drawn. The overlap with the fundamental mode is an effective low-pass filtering in the spatial-frequency domain, with a cutoff frequency of approximately $|f_x| = \frac{1}{d}$, as is expected from the finite NA of the fundamental mode.

However, since the filtered fields, $\tilde{E}_{signal}(f_x) \cdot \tilde{M}_{1/d}(f_x)$, is then convolved with a dirac-comb, aliasing occurs (see Supp. Fig. S8). Thus, replicas of the measured signal spatial spectrum will appear in intervals of $\Delta f_x = \frac{1}{p}$. Aliasing of frequencies that are well sampled (fulfilling the Nyquist sampling rate, i.e. $|f_x| < \frac{1}{2p}$) appears at frequencies that are higher than $f_{cutoff} = \frac{1}{2p}$, and thus they can be removed by low-pass filtering. The result of such filtering is presented in Supplementary Figure S2(D-F). Note that in the spatial domain, the frequency-domain lowpass filtering is a convolution with a blur kernel of size $2p$, i.e. a Fourier-interpolation of the sampled field (Supplementary Figure S2d).

Note that if \tilde{E}_{signal} contains high spatial frequency components of $|f_x| > \frac{1}{2p}$ that remain after the physical filtering of the fiber mode, they will be aliased to the passband of the lowpass filter ($|f_x| < \frac{1}{2p}$) and will result in image artefacts, which may be reduced by advanced image reconstruction algorithms. Importantly, in our experimental realization such high spatial frequencies are of low amplitude, due to the intentionally sufficiently long minimal distance between the object and the fiber distal facet.

To illustrate the above analysis, Supplementary Figure S8 displays a numerical example for the results of sampling, aliasing, and digital filtering in one dimension.

Figure S8: Fourier filtering (interpolation) process - 1D numerical example. (A) spatial distribution of the fields at the fiber facet: in the spatial domain, the holographically measured field through the MCF, $E_{meas}(x)$ (red) is the result of coupling of the incident field, $E_{signal}(x)$ (dashed cyan) to each of the MCF cores. The coupled field to each core is the overlap of the incident field with the fundamental mode of the core, $M(x)$ (inset, yellow). (B) The Fourier transform of the fields in (A): the spatial sampling by the ordered cores having a pitch, p , results in replicas of the original angular spectrum by spacing of $1/p$. (C) Applying a low pass filter with a cutoff frequency of $f_{cutoff} = \frac{1}{2p}$ removes the spectral replicas aliased to high frequencies. The resulting low-pass filtered field (solid blue) closely resembles the original field angular spectrum (cyan). The remaining difference is the original spatial frequency content at $f_x > f_{cutoff}$ that either was not coupled to the fiber or filtered by the low pass filter, and aliased to $f_x < f_{cutoff}$. In our experiments the high spatial frequencies content is low due to the fiber parameters and illumination/detection geometry. (D) Inverse Fourier transform of the filtered field compared to the original field.

8. -Fig. 1 outlines as parameters of the laser source, 640 nm wavelength, 200 μm coherence length. At page 6, methods, 400 μm coherence length is mentioned. Which value is correct?

We thank the referee for noticing this error in our text. The correct value is a coherence length of 400 μm FWHM, and was corrected in figure 1. To address this point, we have added the supplementary figure S4, presenting the measured coherence function of the laser used in the article (as seen in answer to question 4) .

9. *-Fig 4 shows a high speckle background. Is this to be expected at biological tissue? Can this be resolved? What does the speckle pattern depend on? This should be addressed.*

We thank the referee for pointing out the speckle background present in our results. We have added the following text to the discussion

Speckle noise is present in all results due to the coherent imaging nature of FiDHo, in a similar fashion to wide-field OCT. One reason is the nature of the reconstruction, where even a flat-phase target may introduce speckles: as the hologram signal from two neighboring cores is back-propagated, the non-delta response function of the system causes some spreading of the signal, and the overlap causes the formation of subjective speckle. In addition, multiple reflections and scattering present in biological tissues cause objective speckle formation.

To alleviate some of the speckle noise, similar approaches used in full-field OCT can be utilized. For example, incoherent compounding can be used, where the illumination and reference beam will be scanned over different MCF cores, creating multiple independent reconstructions of the same object. Coherent compounding of such results can improve the imaging resolution. An additional solution is incorporating computational denoising into the reconstruction algorithm e.g. using a compressed sensing approach [36].

10. *-Methods, “At the distal end, the small partially reflecting mirror surface is placed at a distance of $z_m=2$ mm from the fiber distal facet. The partial reflective mirror was fabricated by E-beam evaporation (EBPVD) of a 10nm -thick layer of Titanium on a 1 mm glass slide”. Regarding the ideal placement and dimensioning of the distal reflector several questions came up. As far as I understand, the realized setup employs macroscopic cover slides as spacer and reflector. In a real-world application this would need to be miniaturized to not enlarge the endoscope diameter.*

-10A) How would this be realized and how would this affect the transfer function of the distal optics, for instance due to reflection at the lateral surface?

We thank the referee for raising this point regarding the miniaturization of the distal mirror. To address it in the revised manuscript we detailed an implementation based on a glass cylinder of the same diameter as the fiber to realize a miniaturized distal mirror. In order to present and analyze the potential reflections of this solution, we have added a new section to the Supplementary Materials, which is appended below. In addition, we have added to the main text of the revised manuscript a clarifying statement regarding the fact that the partially-reflecting mirror used in our experiments was a glass cover-slide. The added paragraph in the Discussion Section reads:

“While in our proof-of-principle experiments the distal mirror was held separately from the fiber, it can be incorporated in a miniature tip composed out of a cylinder glass spacer (see Supplementary Section S5 and Supp. Fig S9).”

The new Supplementary Section reads:

Section S5: Distal Mirror Design

While to minimize footprint, the distal partially-reflecting mirror should ideally have a diameter not larger than the fiber diameter, in our proof-of-principle experiments the distal mirror was realized by a glass spacer and reflector, both made of two glass slides (with the

reflector being coated at one facet). These had a diameter that was larger than the diameter of the fiber. However, a miniaturized version of the distal mirror can be realized by attaching a glass cylinder having the same diameter as the fiber to the fiber distal end, maintaining a minimal footprint. We display this design in Figure S9, which will be at the focus of future work. While such a distal mirror design is straightforward to implement, it can introduce additional spurious reflections from the lateral surfaces of the cylinder. Note that such spurious reflections are expected to be relatively weak due to the small index contrast between the cylindrical spacer and the surrounding medium. In addition, since light is both emitted and collected by a fundamental mode of the fiber cores, having a limited NA, the amplitude of the collected signal in the fundamental mode that results from reflections from the lateral surface of the cylindrical spacer is reduced twice: once in illumination and once in collection, as coupling back to the core depends on the angle of incidence.

Nonetheless, several solutions can be used to mitigate and further reduce such reflections: An anti-reflective coating can be applied to the lateral surface of the spacer. Alternatively, an absorptive/diffusive surface or a varying radial refractive index can be used to lower such reflections. In addition, a sufficiently short length for the cylindrical spacer length can be chosen, to ensure that higher angled beams do not reach the lateral surfaces (Fig.S9 dotted curve), at the price of a smaller imaging FOV.

Finally, if required, a computational approach that takes into account the total reflected reference field, including the residual spurious reflections, can be implemented in the reconstruction process.

Fig. S9. Geometrical sketch of a proposed miniaturized partially-reflecting distal mirror composed of a transparent cylinder of diameter equal to the fiber diameter, D , and length Z_m . The reference beam cone marked in red, depicts the propagation of the fundamental mode, where most energy lies in angles limited by its numerical aperture: $NA = n \cdot \sin\theta_1$. The mirror distance, Z_m , is optimally chosen such that the reflected reference beam covers the entire fiber facet. The red dashed lines illustrate the path of higher NA illumination, emitted via higher fiber modes or as found outside the FWHM of the fundamental mode. The dotted line illustrates a spurious, unwanted, weakly reflected reference illumination, which can be suppressed by proper choice of spacer materials and/or coating. θ_{max} depicts the maximum angle of incidence of detected light originating from objects points at the edge of the FOV: $\tan(\theta_{max}) \leq \frac{D}{2Z_m} + \tan(\theta_1)$.

10B) 2 mm distance of the partial reflecting mirror and 270 μm diameter of the fiber results in very small triangulation angles, therefore low axial resolution and signal-to-noise-ratio (SNR). What is the achieved axial resolution and the SNR?

We thank for referee for pointing out the unclarity in this issue. The fiber used in our article is mostly the Schott fiber, of diameter 1.1mm, where only $D = 650\mu\text{m}$ was used, and the Fujikura fiber was only used in the supplementary figure S4. We have corrected the last paragraph in Section S2 fiber parameters:

The fiber and laser used throughout this article is a subset of Schott 1533385, $D = 1100\mu\text{m}$, $NA_{\text{core}} \sim 0.2$, $\lambda = 640\text{nm}$ where only $D = 650\mu\text{m}$ was used. In all experiments, a mirror distance of $z_m = 2000\mu\text{m}$ was used. An additional fiber, Fukijura FIGH-06-300S, $D = 270\mu\text{m}$, $NA_{\text{core}} \sim 0.3$, was used only in figure S4.

The required distance to the distal mirror, is $Z_m > \frac{D}{4NA_1}$ as explained in section S2. However, we wish to thank the referee again for drawing our attention to a mistake found in the formula, missing the refractive index. The correct formula should be $Z_m > \frac{nD}{4NA_1}$. We have added the following calculation and figure to the supplementary section S2: Choice of optimal system parameters.:

For the Schott fiber used throughout the article, $D=500\mu\text{m}$, $NA_1=0.2$, used in optical oil the required distance is $Z_m > 1200\text{mm}$.

Finally, we have also added the supplementary figure S9 (see above) to the Supplementary Materials to help explain the effects of moving the distal mirror.

11. -How to discriminate signals of scattering in blood and water layers from the tissue surface, if the coherence gating cannot be used (scattering layers smaller than the coherence length)?

Indeed, as in OCT, the axial discrimination is determined by the coherence length. We note that the coherence length used in our experiments is not the minimal coherence length that can be exploited in FiDHo. For our experimental setup (MCF and distal mirror parameters) one can utilize a coherence length as short as ~20 microns that will still allow interference over the entire distal facet of the fiber. The lower bound for the coherence length is derived from geometric considerations, assuring that the reflected signals from the entire FOV will interfere with the reference signal, over the entire distal facet of the fiber, i.e. by verifying that the path differences are smaller than the coherence length. We mention that such a coherence length is of the order of the size of a single red blood cell. To note this in the revised manuscript, we have added the following statement:

A shorter coherence-length source would be beneficial to both improve the axial-sectioning resolution and reduce the minimal working distance. The shortest coherence length that can be used in FiDHo that provides interference in all fiber cores is determined by the fiber diameter and distal mirror distance. For the experimental parameters of the presented system the coherence length can be as short as ~20μm.”

12. -The described optical setup on the proximal side is highly complex considering the primary task is to realize two spatially coinciding and temporally separated optical beams, only. The authors give sound reason regarding he additional components such a balancing of

intensity and suppression of reflexes. The authors should discuss the influence of imbalanced intensity and reflections onto the image quality in more detail.

The setup for used for realizing and developing our approach is indeed more complicated than needed for a simple implementation, which can use a balanced fiber-based setup. To note this fact we have included the following clarification in the original manuscript Methods section:

To allow maximal flexibility in optical alignment and tuning of the various parameters, such as power splitting, the experimental setup used for our proof of principle experiments is based on a rather large number of bulk optical components that allow maximal flexibility in optical alignment and tuning of the various parameters, such as power splitting. A simpler, more applicable design can be realized by using a single-mode fiber (SMF) based balanced interferometer design.

We kindly note that intensities balancing is extensively analyzed in the supplementary sections “Distal mirror reflectively and reference- and signal-arms powers ratio”. And the suppression of spurious reflections is addressed in the same section.

Reviewer #2 :

The paper from Badt and Katz reports on an interesting scheme to achieve amplitude and phase resolved refractive index imaging through a fiber optic bundle. Using low coherence full-field phase shifting interferometry is not new and its implementation has been performed through optical fiber. What is novel here is its implementation through a fiber bundle to perform wide field imaging. Although the interest of such technique might remain limited (due essentially to the poor image quality) I think the idea is novel enough such as its may be published. The paper is clear and well written and the supplementary data are very helpful to discuss the various important parameters and limitations of the proposed technique.

We thank the referee for his kind words and the observations made on our approach. We agree that the interest might be limited in the current image quality, however major improvements can be easily made and our paper merely presents a proof-of-concepts. We have taken the referee's advices and added a full section discussing the digital filtering, as well as suggestions for improving image quality.

There are few points the authors could address in a revised version of the manuscript to improve its clarity.

1. *Reconstruction process: It is said that E_{illum} is assumed to be a Gaussian beam, however the fiber bundle used by the authors are known to have bi-modal structures in some cores. Did the authors observed cases where the reconstruction was odd and did they post-select results coming from reference fields emerging from a single mode fiber core?*

We thank the referee for raising this important point that was not addressed in detail in our original manuscript. Indeed, each core of the MCF supports multiple modes. We wish to emphasize that no post selection was done, but measurement of only the fundamental mode illumination in our holographic process was assured by the short coherence temporal gating, as the arrival time of the higher order modes is larger than that of the fundamental mode.

While higher-order modes appear in the detection step, we attribute these to higher angle of reflections from the object, and potential modal coupling in the back propagation from the distal end to the proximal end. Since these occur at the desired time-gate *after* the holographic interference, their power is still useful for the object reconstruction, since in the detection process, the MCF serves as merely a light intensity relay. Following a similar question by referee 1 regarding the higher modes in the illumination/reference step, we have revised our manuscript to address this point by adding a detailed discussion to both the Discussion and the Supplementary section, including new experimental results to the revised manuscript. We kindly refer to our answer to question 1 of Referee 1.

2. *The digital filtering of the MCF pixilation and its effect on imaging is not very clear to me. (1) Could the author rephrase this key sentence "by Fourier-interpolating the holographically measured fields, effectively interpolating the fields between the cores at the fiber facet". Also what are the effect on the imaging to filter out the replica, I guess there is a loss of brightness that leads to dimmer contrast? In general, it would be valuable to discuss more deeply the effects of this digital filtering.*

We thank the referee for drawing our attention to the insufficient clarity and depth of discussion of the original manuscript's on this subject. To address this point we have rewritten the Methods section "Digital filtering of MCF pixilation" and have added a new Supplementary Section that discusses in depth the MCF sampling artefacts and their physical and digital suppression, including a one-dimensional theoretical analysis and numerical example.

The revised Methods section reads:

Digital filtering of MCF pixilation

Due to the low fill-factor of the MCF cores, the holographically-recorded field at the fiber facet plane is pixelated, and is effectively under-sampled by the MCF cores (**Error! Reference source not found.** B, inset). This under-sampling manifests itself as aliasing in the angular frequency domain, which results in "ghosts" replicas in the back-propagated reconstructed images (Fig. S2 C). These artefacts can be suppressed by digitally filtering the reconstructed fields, as explained in detail in Supplementary Section S3. Briefly, we apply a low-pass filter on the sampled fields before back-propagation (Fig.S2). This effectively performs a Fourier-interpolation of the holographically measured fields between the cores at the fiber facet (Fig. S2 D). We set the cutoff frequency of the low-pass filter to filter the aliased spatial frequencies above $k_{cutoff} \approx \pi/p$, where p is the core-to-core pitch. The Fourier interpolation also effectively limits the detection NA, which dictates the reconstructed fields resolution. Similar to other works employing MCFs for imaging, since the NA of the fundamental mode in each core is larger than $k_{cutoff}/2\pi$, some aliasing artefacts will always be present if ordered-cores MCFs are used, and may be avoided by using MCFs with aperiodic arrangement of cores [46]. In FidHO, the aliasing artefacts are naturally suppressed by the minimal value of the object distance from the fiber facet, set by the reference mirror distance. The minimal object-distance and the fact that the illumination is provided by the fundamental mode of a single core leads to an inherent limit on the maximal angle (NA) of the reflected fields that at the fiber facet (see Supp.Fig. S9). For an in depth discussion of digital filtering, please refer to Supplementary Section S3.

The presented results were obtained using an MCF with identical cores arranged on an ordered grid with very low crosstalk (Schott 153385). We obtained similar results using an MCF with inhomogeneous cores arranged on an imperfectly ordered grid (Fujikura FIGH-06-300S) (Fig. S4). In this case of imperfectly ordered grid, the spatial cutoff frequency of the Fourier interpolation filter was set according to the average pixel pitch. A discussion on the effects of the various fiber parameters is presented in Supplementary section S2.

The new Supplementary Section is referred to in the main manuscript text by the following paragraph:

"The MCF pixilation at the fiber facet, which may introduce ghosting due to aliasing in k-space, is removed by digital filtering, and aliasing is suppressed by the illumination and detection geometry set by the reference mirror distance (see "Digital filtering of MCF pixilation" in Methods and Supplementary section).

An additional reference to the detailed discussion was also added to the "Resolution and field of view" chapter in the main text, which reads:

"...The resolution is diffraction-limited by the numerical aperture (NA) of the recorded holograms, which is dictated by the NA of the fundamental mode used for illumination and detection, and by the digital filtering of the holograms at $f_{cutoff} = 1/2p$, that is performed to filter aliasing artefacts caused by pixelation (sampling) of the ordered MCF cores, where p is the core-to-core pitch (see Methods and Supplementary S3)."

The new Supplementary Section reads:

Section S3: Digital filtering of MCF pixilation

The distal holography approach is based on holographic recording of the field at the distal facet. However, since the field hologram is not sensed directly but is effectively sampled the fiber cores, having a limited fill-factor due to the core-to-core pitch being larger than the mode diameter, sampling-induced artefacts (resulting from aliasing in spatial frequency space), appear in the reconstructed field. To achieve the high image quality of FiDHo, these sampling-induced artefacts need to be suppressed. Here, we obtained the required suppression of these artefacts by a combination of imaging and illumination geometry and natural filtering of the MCF cores, which attenuated high spatial frequencies, in addition to digital filtering in post-processing. In this section we provide a mathematical and numerical analysis of the sources for the sampling-induced artefacts, and of the effectiveness of the approaches we have implemented to suppress them.

We first briefly present and explain the experimentally observed artefacts, as displayed in Supplementary Fig.S2a-c. In our experiments with a commercial MCFs produced by Schott, cores are located on a hexagonal grid with a core-to-core pitch, $p = 8\mu\text{m}$ (Fig.2A). Thus, the incident field hologram is effectively sampled at a spatial sampling frequency, $f_s = 1/p$. As result of the convolution theorem, in the Fourier (angular frequency) domain, this spatial domain sampling results in generation of replicas of the object angular spectrum at shifts of $\Delta f = m f_s$, where $m = \pm 1, \pm 2, \dots$ (Fig.S2B). If the object plane is located at a distance, z , which is effectively at the far-field of the MCF, these aliased replicas will limit the FoV to $\sim \lambda z/p$ [46]. If the object is located at shorter distances from the fiber, the angular frequency artefacts will results in object replicas inside the imaged FoV (Fig.S3C).

To yield the following mathematical analysis more accessible and without loss of generality, we perform the analysis in one dimension (1D). Extension to two dimensions (2D) is straightforward.

Consider an electric field, $E_{\text{signal}}(x)$, being measured through a 1D array of waveguides spaced evenly with a pitch p , representing the MCF cores. Each waveguide can guide light with a complex amplitude that is proportional to the overlap-integral of the electric field that impinges upon it with a certain mode M_d , of typical mode-field-diameter d . Therefore, the measured field on the other side of the array will be:

$$E_{\text{meas}}(x) = \left((E_{\text{signal}}(x) * M_d(x)) \cdot \text{comb}_p(x) \right) * M_d(x)$$

Where $*$ symbols a convolution that realizes the overlap integral with the fiber mode, comb_p is the Dirac-comb composed of delta-functions spaced at intervals of p .

In the angular frequency domain, this sampled field is given by:

$$\tilde{E}_{\text{meas}}(f_x) = \left(\left(\tilde{E}_{\text{signal}}(f_x) \cdot \tilde{M}_{1/d}(f_x) \right) * \text{comb}_{1/p}(f_x) \right) \cdot \tilde{M}_{1/d}(f_x)$$

Where the convolution theorem was used. Assuming that M_d is a gaussian-like function of typical width d we can assume $\tilde{M}_{1/d}$ to be of typical width $\frac{1}{d}$. We note that in an MCF the core diameter is smaller than the core spacing: $d < p$.

Several conclusions can be drawn. The overlap with the fundamental mode is an effective low-pass filtering in the spatial-frequency domain, with a cutoff frequency of approximately $|f_x| = \frac{1}{d}$, as is expected from the finite NA of the fundamental mode.

However, since the filtered fields, $\tilde{E}_{signal}(f_x) \cdot \tilde{M}_{1/d}(f_x)$, is then convolved with a dirac-comb, aliasing occurs (see Supp. Fig. S8). Thus, replicas of the measured signal spatial spectrum will appear in intervals of $\Delta f_x = \frac{1}{p}$. Aliasing of frequencies that are well sampled (fulfilling the Nyquist sampling rate, i.e. $|f_x| < \frac{1}{2p}$) appears at frequencies that are higher than $f_{cutoff} = \frac{1}{2p}$, and thus they can be removed by low-pass filtering. The result of such filtering is presented in Supplementary Figure S2(D-F). Note that in the spatial domain, the frequency-domain lowpass filtering is a convolution with a blur kernel of size $2p$, i.e. a Fourier-interpolation of the sampled field (Supplementary Figure S2d).

Note that if \tilde{E}_{signal} contains high spatial frequency components of $|f_x| > \frac{1}{2p}$ that remain after the physical filtering of the fiber mode, they will be aliased to the passband of the lowpass filter ($|f_x| < \frac{1}{2p}$) and will result in image artefacts, which may be reduced by advanced image reconstruction algorithms. Importantly, in our experimental realization such high spatial frequencies are of low amplitude, due to the intentionally sufficiently long minimal distance between the object and the fiber distal facet.

To illustrate the above analysis, Supplementary Figure S8 displays a numerical example for the results of sampling, aliasing, and digital filtering in one dimension.

Figure S8: Fourier filtering (interpolation) process - 1D numerical example. (A) spatial distribution of the fields at the fiber facet: in the spatial domain, the holographically measured field through the MCF, $E_{meas}(x)$ (red) is the result of coupling of the incident field, $E_{signal}(x)$ (dashed cyan) to each of the MCF cores. The coupled field to each core is the overlap of the incident field with the fundamental mode of the core, $M(x)$ (inset, yellow). (B) The Fourier transform of the fields in (A): the spatial sampling by the ordered cores having a pitch, p , results in replicas of the original angular spectrum by spacing of $1/p$. (C) Applying a low pass filter with a cutoff frequency of $f_{cutoff} = \frac{1}{2p}$ removes the spectral replicas aliased to high frequencies. The resulting low-pass filtered field (solid blue) closely resembles the original field angular spectrum (cyan). The remaining difference is the original spatial frequency content at $f_x > f_{cutoff}$ that either was not coupled to the fiber or filtered by the low pass filter, and aliased to $f_x < f_{cutoff}$. In our experiments the high spatial frequencies content is low due to the fiber parameters and illumination/detection geometry. (D) Inverse Fourier transform of the filtered field compared to the original field.

To better address this point and provide more detailed information on the filtering/interpolation process, we have revised Supplementary Fig.S2 and its caption, which now read:

Fig. S2. Fourier Interpolation on ordered MCF (A) The measured distal field on a Schott fiber used throughout the article, shows the individual cores with a constant inter-core pitch, p , which perform effective spatial sampling of the field at a spatial frequency of $1/p$. (B) In the spatial frequency domain, replicas appear as result of the spatial sampling, as expected from the convolution theorem. The dashed red circle marks the spatial frequency range that is within the Nyquist sampling criterion ($|f| < \frac{1}{2p}$, i.e. $|k| < \frac{\pi}{p}$). (C) Reconstruction of the object directly from the measured field of (A), displays ghost replicas, due to the frequency-space aliasing. (D, E, F) show the same data, after low-pass filtering the aliased high spatial frequencies outside the dashed circle in (B). Scale bars: A, C - $100_{\mu m}$ B - $\frac{2\pi}{p}$

3. In Fig 4 the cell can be viewed only in the phase contrast, this is quite limiting for the applications and this is related to the minimal reflectivity that the sample must produce to make the technique viable. Could the authors discuss what would be possible to achieve to improve the minimal reflectivity that the FiDHo technique can achieve.

We thank the referee for pointing out the need for further discussion on image contrast and minimal detected reflectivity. To address this point, we have added two discussions to the main manuscript text. In short, the fundamental limits of the imaging contrast-to-noise are equivalent to those of OCT, and include the power ratio between the reference and illumination arm, the number of frames used for phase-shifting, and the camera well-depth. These are analyzed in-depth for OCT in reference [41] (also [41] of the original manuscript), as well as in [“Full-field optical coherence microscopy”, E. Beaurepaire, A. C. Boccara, M. Lebec, L. Blanchot, and H. Saint-Jalmes, Optics Letters 23, 4, 244-246 (1998)]. The contrast can also be improved by using a shorter coherence length source, which would be able to distinguish between the different reflecting layers (e.g. the cell walls and the microscope slide), which was not possible with the coherence-length of the laser source used in our experiments.

The added discussion paragraph in the revised manuscript also includes a reference to “Full-field optical coherence microscopy”, as reference [40]. It reads:

The image contrast of FiDHo is a coherent reflection contrast, and has the same origins and limitations as in OCT. The image contrast and contrast-to-noise limits are analyzed in depth for OCT in [40,41]. In short, the image contrast-to-noise is dictated by the target reflectivity, index contrast, tissue scattering, laser coherence length, number of frames used for phase-shifting, and the camera pixel well-depth. In addition to these conventional limits, FiDHo imaging contrast also depends on the spurious reflections present in the system, inter-core crosstalk, and the applied digital Fourier filtering (see Methods).

We analyze the optimal reference wave relative-intensity and mirror reflectivity that would maximize the signal to noise of FiDHo measurements in Supp.Section S2. This section was referred to in the original main text in the following statement:

“For optimal imaging performance the parameters of the MCF, distal-mirror, and powers ratio of the object- and reference-arm should be set according to several considerations. These considerations are analyzed in detail in section S2.”

Finally, we have rephrased the statement on potential improvement of the presented system:

The signal to noise and minimal detected reflectivity can be improved by using cameras with higher frame-rates and larger well-depth, as well as shorter coherence length sources [41].

4. *Fig 6 on the chicken breast is not informative at all, rather it put the technique down for useful imaging. Could the author image a more interesting 3D sample (a mm insect, a pollen grain....)*

We thank the referee for this comment. Our motivation for originally including Fig.6B was to demonstrate depth-resolved imaging of a large FoV via mosaicking/stitching. To address this point we have replaced figure 6B with a new experimental result obtained from a resolution target inclined at an angle with respect to the distal fiber end. The revised figure and figure caption are:

Figure 6: Imaging three-dimensional objects. (A) A 3D image of a target composed of two stacked resolution targets with a spacing of $\sim 300\mu\text{m}$ reconstructed by super-posing two reconstructed images acquired with two appropriate time delays, $\tau = 2z_0/c$, with $z_0 = 1030\mu\text{m}$ (pink) and $z_0 = 1320\mu\text{m}$ (yellow). (B) A tilted USAF reflectivity target, reconstructed by stitching 20×20 sub-images, each with the fiber field of view given by the dashed square. The varying depth of each sub-image is retrieved from the time-delay scan. The target was placed at an angle w.r.t the distal tip to demonstrate 3D imaging capabilities. Scale bars: A - $50\mu\text{m}$ B - $500\mu\text{m}$

We agree with the referee that an example of three-dimensional imaging of a biological sample would be a powerful demonstration of FiDHo. After much efforts towards this goal using our current experimental system, we have concluded that the $400\mu\text{m}$ coherence-length of our laser source is too long and sub-optimal for this task. Unfortunately, while many efforts were made to include the optimal results in the revised manuscript, such an optimal shorter coherence-length source is not available to us at this time, and the demonstration of high axial resolution 3D imaging of biological sample will have to be left for a future study. To address this point we have added the following statement to the revised manuscript Discussion:

We have provided a proof-of-principle demonstration and in-depth investigation of a novel endoscopic imaging technique based on coherent elastic scattering contrast. As in OCT, such endogenous reflection contrast can be used for diagnosis of various biomedical parameters, e.g. the detection of abnormal cells in the gastrointestinal tract [44]. Such a study requires the use of shorter coherence-length source and potentially higher well-depth cameras and will be at the focus of future work.

Where [44] refers to “Criteria for the diagnosis of dysplasia by endoscopic optical coherence tomography” Pfau et al. *Gastrointestinal endoscopy* 58.2 (2003): 196-202”.

5. *Minor: the acronym FiDHo is not defined in the text (only in the title)*

We have added the following definition to the main text, at the end of the introduction:

“Here we present Fiber Bundle Distal Holography (FiDHo):”

Reviewer #3

The work by Noam Badt and Ori Katz presents an unpixelated video-rate micro-endoscopy imaging technique that does not require bulky optics at the distal end, is insensitive to bending distortions, capable of axial sectioning and with diffraction limited lateral resolution over an extended distance behind the distal fibre facet/partially reflective mirror. I find the work highly original, with an intriguing set of imaging characteristics not matched by any other available micro-endoscopy approach. Furthermore, the presented method has, in my opinion, a strong potential to be quickly adopted in a clinical setting due to its relative simplicity.

The paper is very well written, the principles of the method are clearly explained and the claims are supported by experimental evidence. I would also like to thank the authors for providing useful information for someone who would like to reproduce the work, including thorough discussion of unwanted back-reflections in the system, optimal reflectivity of the distal mirror and the optical setup design and sample considerations.

We sincerely thank the referee for the kind words.

Based on the above I recommend the manuscript to be published in Nature Comm. However, I have several comments that I think would further improve the quality of the manuscript if addressed:

1. *The claim of 3D imaging, and even “3D diffraction-limited” (page 5, line 183) is not the best choice of words given the current data. The system has axial sectioning capability, with experimental evidence hinting at 300um axial sectioning resolution. That is not 3D diffraction limited. I also think a more thorough discussion of the axial sectioning resolution achievable with the system is in order. At the moment, the reader has to look up coherence length of the laser in the text and make estimates of axial sectioning capability from such scattered information. I would like to see a transparent discussion of axial sectioning resolution from the authors.*

We thank the referee for pointing on the inaccurate choice of wording describing the imaging resolution, and the required additional discussion and clarification regarding the axial-resolution, which was also raised by Referee 1.

To address the choice of wording we have first changed the phrasing of the relevant sentence from:

“These include video-rate 3D diffraction-limited, axially sectioned label-free imaging...”

To

“These include video-rate imaging with a transverse diffraction-limited resolution, axially-sectioned label-free imaging...”

Following this we have added the following discussion and analysis to the “Results” section of the main text, under the subtitle “Resolution and Field-of-View”, and have added a measurement of the coherence length of the laser used as a new Supplementary Fig. S7. The additional discussion reads:

The axial resolution of FidHO is determined by two factors: the axial sectioning due to coherence gating, and axial resolution due to the numerical aperture of the imaging system. First, similar to OCT, the coherence gating yields an axial sectioning with an axial resolution dictated by the coherence length of the laser used, l_c . In our experiments $l_c = 400\mu m$. However, a shorter coherence length source can be used to improve the axial sectioning resolution. An experimental measurement of the coherence-gating axial sectioning in our experiments is presented in Supp.Fig. S7.

As in other holographic imaging approaches, in addition to the coherence-gating axial resolution, the imaging axial resolution that originate from the NA of the recorded holograms is dependent on the imaged object: for a point-reflector, the axial resolution would be $\delta z \approx \frac{\lambda}{NA^2} \approx 30\mu m$. For a flat-phased object of diameter d_{obj} , $\delta z \approx \frac{d_{obj}^2 + (\frac{\lambda}{NA})^2}{\lambda}$. The axial and lateral resolutions can be improved by advanced computational reconstruction schemes [36] (see Discussion).

Fig S7 Measured Coherence function of the laser source. Total energy of phase-shifted hologram vs. the relative delay between the illumination and reference arms. The measurement was performed on the two beams directly at the output of the interferometer. The measured full-width at half max (FWHM) is $400\mu m \pm 50\mu m$ (red).

2. *Related to point 1. There is discrepancy between coherence length of the laser $l_c = 400\mu m$ (page 6, line 216) in the text and $200\mu m$ in Fig. 1, which makes the axial sectioning capability even harder to extract from the paper.*

We thank the referee for pointing on this inconsistency in our text. The correct value is a coherence length of $400\mu m$ FWHM. We have corrected the value wrongly stated in the original figure 1.

3. How miniature is the partially reflective mirror? Based on information in the Methods, it looks like the miniature mirror is made of two 1mm thick glass-slides, with immersion oil between the facet and the glass slides. I think that since the manuscript repeatedly claims how other methods suffer from bulky optics, more care should have been given to this issue. From my point of view, I think the partial mirror lateral extent can indeed have the diameter of the MCF – the method would work with that, and I do not mind the 2mm extra distance needed between MCF and mirror plus additional 2mm for glass slides. That can all become part of MCF at some point. However, all the current experiments were done with bulky glass slides if I am not mistaken. I understand that this is just a demonstration and making a miniature mirror is a purely engineering problem. However, it should be clear that the experiment was done with bulky glass slides and a hint of a pathway how to make a small add-on on MCF should be part of this paper.

The referee is correct regarding the bulky nature of the partially reflecting mirror. This was indeed a result of our experiments being proof-of-concept experiments, and was not sufficiently emphasized in the original manuscript. The design, fabrication, and assembly of a miniaturized mirror must be addressed to render the approach applicable.

In order to emphasize that our proof-of-principle experiments were all performed with a non-miniaturized glass-slide based mirror, we have added the following clarification to the revised manuscript Discussion:

“While in our proof-of-principle experiments the distal mirror was held separately from the fiber, it can be incorporated in a miniature tip composed out of a cylinder glass spacer (see Supplementary Section S5 and Supp. Fig S9).”

In addition, to address possible miniaturization, we have added a new section to the Supplementary materials, where we propose a miniaturized solution based on a coated glass cylinder, including analysis of potential unwanted reflections from its side surfaces. This new Supplementary Section reads:

Section S5: Distal Mirror Design

While to minimize footprint, the distal partially-reflecting mirror should ideally have a diameter not larger than the fiber diameter, in our proof-of-principle experiments the distal mirror was realized by a glass spacer and reflector, both made of two glass slides (with the reflector being coated at one facet). These had a diameter that was larger than the diameter of the fiber. However, a miniaturized version of the distal mirror can be realized by attaching a glass cylinder having the same diameter as the fiber to the fiber distal end, maintaining a minimal footprint. We display this design in Figure S9, which will be at the focus of future work. While such a distal mirror design is straightforward to implement, it can introduce additional spurious reflections from the lateral surfaces of the cylinder. Note that such spurious reflections are expected to be relatively weak due to the small index contrast between the cylindrical spacer and the surrounding medium. In addition, since light is both emitted and collected by a fundamental mode of the fiber cores, having a limited NA, the amplitude of the collected signal in the fundamental mode that results from reflections from the lateral surface of the cylindrical spacer is reduced twice: once in illumination and once in collection, as coupling back to the core depends on the angle of incidence.

Nonetheless, several solutions can be used to mitigate and further reduce such reflections: An anti-reflective coating can be applied to the lateral surface of the spacer. Alternatively, an absorptive/diffusive surface or a varying radial refractive index can be used to lower such reflections. In addition, a sufficiently short length for the cylindrical spacer length can be

chosen, to ensure that higher angled beams do not reach the lateral surfaces (Fig.S9 dotted curve), at the price of a smaller imaging FOV.

Finally, if required, a computational approach that takes into account the total reflected reference field, including the residual spurious reflections, can be implemented in the reconstruction process.

Fig. S9. Geometrical sketch of a proposed miniaturized partially-reflecting distal mirror composed of a transparent cylinder of diameter equal to the fiber diameter, D , and length Z_m . The reference beam cone marked in red, depicts the propagation of the fundamental mode, where most energy lies in angles limited by its numerical aperture: $NA = n \cdot \sin\theta_1$. The mirror distance, Z_m , is optimally chosen such that the reflected reference beam covers the entire fiber facet. The red dashed lines illustrate the path of higher NA illumination, emitted via higher fiber modes or as found outside the FWHM of the fundamental mode. The dotted line illustrates a spurious, unwanted, weakly reflected reference illumination, which can be suppressed by proper choice of spacer materials and/or coating. θ_{max} depicts the maximum angle of incidence of detected light originating from objects points at the edge of the FOV: $\tan(\theta_{max}) \leq \frac{D}{2Z_m} + \tan(\theta_1)$.

4. Page 3, Line 114, equation (2). It would be beneficial to mention that the propagator can deal with different materials along the way, immersion oil/air etc.

We thank the referee for pointing out on this interesting point. We note that in addition to different homogeneous materials, the propagator can in principle computationally correct for aberrations, as was demonstrated in OCT by the group of Boppart and others. To incorporate these interesting points, the following line was added to the text:

The propagator, $\mathcal{P}_{-z_{prop}}$ can be implemented to take into account changes of refractive index, as well as incorporate computational aberration correction [35].

Where reference [35] is [Adie et al. PNAS 109 (19) 7175-7180 (2012)].

5. Page 4, Line 124, correct to “at the different axial plane”

We have corrected this point in the revised manuscript.

6. Page 4, Line 144, can you explicitly write the z_{obj} so that the end-user can get a feeling for the maximum distance over which the resolution and FOV do not change?

We have changed the phrasing of this sentence to:

The resolution and FoV are independent of imaging distance for distances of $z_{obj} < nD_{fiber}/2NA - z_m$, where D_{fiber} is the MCF diameter, n is the index of refraction of the medium, and NA is the effective numerical aperture of the FiDHo imaging system. In the presented experimental system: $D_{fiber} = 600\mu m, z_m = 2mm, n = 1.51, NA \approx 0.15$, gives a constant resolution $z_{obj} \approx 1mm$.

7. *I would move the Phase-contrast imaging paragraph at the very end of Result section.*

We thank the referee for this suggestion and have given considerate thought to this issue. We note that since the phase shifting capabilities are attributed to a *single, two-dimensional* reconstruction of an object, we found it is beneficial to present these before presenting more complex multi-frame qualities of our system. Therefore, we have decided to maintain the existing paragraph order.

8. *Page 4, Line 167 correct to "collected through"*

We have corrected this point in the revised manuscript.

9. *Is there a way to quantify how insensitive the imaging is to bending?*

Quantifying bend sensitivity is a challenging measurement, as precise dynamic control of the bending requires additional computer-controlled motorized stages, with a specific, well-defined geometry. Unfortunately, we have not incorporated such an apparatus in our experimental setup. However, as can be observed in the Supplementary movie, while dynamically varying bending within the few phase-shifting frames may impact the image quality, once the bending is static, high fidelity imaging is quickly regained, and any static bending or sufficiently slow bending is acceptable. To note this point, we have added the following statement to the main manuscript text discussing the sensitivity to bending:

As can be observed in Supplementary Movie S2, while fast dynamic bending within the three phase-shifting frames may impact the imaging quality, the approach is insensitive to fiber orientation.

10. *Fig 2A. I see LP11 like modes in the inset. Why is the light coupled as an LP11 mode? Can the presence of LP11 influence the reconstruction of the object field?*

The light is coupled to higher modes due to the higher angles of diffraction from the object itself. Since we integrate over each core before we reconstruct, as well as pass the data through a low-pass filter, this does not affect the reconstruction. Nevertheless, the higher modes can effect the illumination and reference arms, and we have added a section addressing these higher modes. The added section is presented below as an answer to question 12.

11. *Fig.1A the units of laser are as subscripts*

fixed

12. *Page 5, line 192. You claim bend sensitivity is minimised by using single mode fibers, yet Fig 2A shows your fibres are multimode and couple LP11 in some cases. Can you please discuss?*

We thank the referee for raising this point. Indeed, the bend sensitivity is minimized when both the illumination and reference beams travel in the same fiber mode. Thus, any fiber bending will result in the same phase distortions imposed on the two beams, and will have no effect on their interference forming the measured holograms of FiDHo. Nonetheless, we chose to demonstrate FiDHo with commercially available MCFs, which are all composed of cores that support multiple transverse, to show that the technique does not require specially-drawn or designed fibers. The reason that using such off-the-shelf MCFs still displays low sensitivity to

bending is due to the combination of: (a) careful effort made to excite only the fundamental mode with both beams; (b) the use of a short coherence length source to time-gate the holographic measured fields to the propagation time of the fundamental mode, effectively filtering the higher order modes (see new Supplementary Fig. S5); and (c) the relatively low modal coupling in the fibers used.

To discuss this point, which was raised also by referee #1 we have added a short discussion following line 192, and an in-depth discussion accompanied by new numerical simulations and experimental investigations to Supplementary Section S4. The discussion in the main text reads:

Bend sensitivity can be minimized by choosing an MCF with low inter-core crosstalk and single-mode propagation in each core. This is since multimode propagation or crosstalk between cores will result in a complex and bend-sensitive illumination- and reference-fields when a fiber is bent. Nonetheless, by careful excitation and temporal-gating of the fundamental mode we were able to demonstrate low sensitivity to bending using commercial MCFs with cores that support several transverse modes. An in-depth analysis and quantification of the effects of high-order fiber modes is presented in Supplementary Section S4.

The new Supplementary Section reads:

Section S4: Analysis of the effects of higher-order modes in each core

As each core of the commercial MCFs used in our work supports several transverse modes, in our experiments care was taken to excite only the fundamental mode by the illumination and reference beams. However, such excitation is not perfect, and in addition fiber bending and manufacturing imperfections may couple the beams injected in the fundamental mode to higher modes. This requires additional care in the measurements and analysis to the possible contribution of higher order modes of each core to the image formation. The most important and very effective experimental measure to minimize the contribution of the higher order modes is our use of the short coherence length source to time-gate the interference with the fundamental mode, which temporally arrives before the higher order modes. This was proved to be sufficient to reduce the contribution of higher order modes to a negligible measure. To illustrate this, below, we experimentally characterize the higher order modes in our experiments, and numerically analyze the potential effects of higher order modes on the reconstructed images.

Experimental measurements

To experimentally characterize the temporal delay and relative amplitude of the second transverse fiber mode, we performed a measurement of the total energy of the recorded hologram as the relative delays between the two interferometer arms (the ‘object illumination’ arm and the ‘reference’ arm) is scanned, for a target object placed at a distance of $200\mu\text{m}$ from the distal mirror. The results of this scan are displayed in Fig S5.A.

As expected, the measured scan displays three dominant peaks (i,ii,iii): the most dominant peak (i) appears at zero relative delay, and is dominated by the interference of the strong reflection from the distal mirror in one arm with the same reflection in the second arm. In addition, it contains the interference of the illumination field reflected from the object with the reference field reflected from the object. As we explain below, when a second mode is present, additional mirror-mirror and object-object interferences are contributing to the hologram due to the second mode (see Fig.S5E and explanation below). The second time-gated interference peak (ii) is the desired interference between the fundamental mode illuminating the object and the fundamental mode reference beam reflected from the mirror, used for imaging in our experiments. This peak appears when the delay is set to the object distance: $\tau = 2z_o/(c/n)$.

The third time-gated interference peak (iii) is the result of the reflections from the second mode in the illumination arm with the fundamental mode in the reference arm (see temporal diagram in Fig.S5E).

In accordance, the spatial shape of the holograms measured at these three delays reveal the nature of the different fields that take part in each interference: for (i) the hologram reveals the distal mirror flat-phase reflection (Fig.S5B); for (ii) the hologram reveal the object reflected field on the facet (Fig.S5C), which is used for image reconstruction; for (iii) the hologram reveals the second-order mode illumination of the distal mirror, as measured by the fundamental mode (Fig.S5D). Most importantly, this experiment displays how the short time-gating allows not only to filter the strong mirror reflection, but also to effectively filter the unwanted contributions of the second (and higher) modes.

To clarify the different expected contributions to these three interference peaks (i-iii), we have plotted in Fig.S5E a sketch of the reflections from the distal mirror and object for the first two modes in each arm, for the three different delays (i-iii).

As in Supplementary Section S2, we denote the fields of the reference arm by subscript (R), the fields from the illumination arm by subscript (I), the reflection from the distal mirror or object are denoted by subscripts (m,o) respectively. The first and second mode are denoted by superscript (1) and (2). The fields generated by the reference arm are thus: $E_{R,m} = E_{R,m}^{(1)} + E_{R,m}^{(2)} + E_{R,o}^{(1)} + E_{R,o}^{(2)}$, and the fields generated by the illumination arm are thus: $E_{I,m} = E_{I,m}^{(1)} + E_{I,m}^{(2)} + E_{I,o}^{(1)} + E_{I,o}^{(2)}$. As result of the short time-gate, the fields that contribute to each of the interference peaks, are only those that temporally overlap, as marked by the dashed vertical lines in Fig.S5E. At the desired delay for imaging (ii), the measured hologram is the result of the interference of the fundamental-mode reflection from the distal mirror of the reference beam with the reflection of the fundamental mode of the illumination beam from the object, as desired: $E_{I,o}^{(1)} E_{R,m}^{(1)}$. The relatively weak interference of the second mode illuminating the mirror and object ($E_{I,o}^{(2)} E_{R,m}^{(2)}$) is coherently added to the strong hologram of the first mode. The interference due to the second mode is very small in magnitude since its amplitude is squared in the recorded hologram (both reference and illumination fields are excited weakly in the higher order mode). From the measured experimental trace (Fig.S5A) we estimate that the energy in the second order mode is approximately 2% of the fundamental mode energy.

Fig S5: Effects of higher-order fiber-modes on time-gated holograms. (A) Total phase-shifted hologram energy vs delay introduced between the reference- and illumination-arms for a USAF-target placed at $z_{obj} = 205\mu\text{m}$. Three dominant peaks (i,ii,iii) are observed when the illumination delay matches either the zero relative delay (i), in which the reflected illumination (‘object field’) is dominated by the reflection from the distal mirror, the reconstructed hologram at this delay is a flat-phase and amplitude one (B). The second peak (ii) occurs when the delay matches the object distance $\frac{z_o n}{c}$, in which the object field is dominated by the first mode reflected from the target, and the reconstructed hologram is the object diffraction (C). The third peak (iii) occurs when the delay is equal to the difference between the second and first fiber mode arrival times, in which the distal object field is dominated by the reflection of the second mode from the distal mirror, and the reconstructed hologram is the diffraction of the second mode (D). Sketch (E) depicts the four delayed beams at the distal tip for each of the two arms, considering two fiber modes. Triangles and rectangles represent the mirror and object reflection respectively, solid and dashed lines represent the first and second mode respectively

and red and yellow represent the reference and illumination arm respectively. The three peaks (i,ii,iii) are displayed as three different illumination delays. From the experimentally measured relative energies, we retrieve the approximated values for the reflected intensity from the object: $R_0 = 0.6\%$, and the second mode relative intensity: $M_2 = 2\%$

Numerical investigation of higher-order mode impact on reconstructed images

In the previous section, we have shown that the experimentally measured hologram at the optimal time-delay is the sum of two interference terms: $E_{I,o}^{(1)}E_{R,m}^{(1)} + E_{I,o}^{(2)}E_{R,m}^{(2)}$. The first is the desired hologram generated by the fundamental mode, and the second is a significantly weaker undesired hologram that is generated by the second mode. Experimentally, the weak undesired hologram contribution is difficult to measure and separate from other noise and background contributions. Thus, in order to complement the experimental investigation and characterization of the higher-order modes, we have performed a numerical simulation where we study the impact of the contribution of the second mode on the reconstructed image.

The results of this numerical investigation are displayed in Supplementary Figure S6. The results are obtained from a simulation based on digital angular spectrum propagation, which is performed separately for each mode of the MCF core. The holographic recorded field is simulated in several steps: (i) the illumination field from a selected single mode ($j = 1, 2, \dots$) from a single core at the MCF distal facet is propagated to the object plane; (ii) the field illuminating the object is multiplied by the 2D object reflection function; (iii) the reflected field is propagated back to the MCF distal facet to produce the object field from the illumination arm at the distal facet for each mode, j : $E_{I,o}^{(j)}(x, y)$; (iv) the same steps are performed for all the fiber modes studied when replacing the object reflection function and distance by the distal mirror reflection, to simulate the reference field at the distal facet for each mode: $E_{R,m}^{(j)}(x, y)$. The simulated holographically measured fields are then calculated for each mode by:

$$E_{holo}^{(j)}(x, y) = \left(E_{R,m}^{(j)}(x, y) \right)^* E_{I,o}^{(j)}(x, y)$$

Following (Eq.1), the reconstructed object field image is produced by normalizing the holographic recorded field by the expected diffraction of the reference mode $\left(E_{R,m}^{(1)}(x, y) \right)^*$, followed by back-propagation:

$$O^{(j)}(x, y, z_{prop}) \propto \mathcal{P}_{-z_{prop}} \left(\frac{\left(E_{R,m}^{(j)}(x, y) \right)^* E_{I,o}^{(j)}(x, y)}{\left(E_{R,m}^{(1)}(x, y) \right)^*} \right)$$

For the first mode contribution ($j=1$), the normalization is correct, and the reconstructed image is the same as in the ideal single-mode case. For other modes, the normalization does not correctly compensate for the contribution of the reference mode, and the holographic recorded

field is multiplied by an erroneous factor of: $\frac{\left(E_{R,m}^{(j)}(x, y) \right)^*}{\left(E_{R,m}^{(1)}(x, y) \right)^*}$ (in addition to a different illumination of the object), which results in imaging artefacts in the reconstruction fields that are a coherent sum of the different modes contribution: $O_{reconst}(x, y) = \sum_j O^{(j)}(x, y, z_{prop})$.

Fig. S6 displays simulated reconstructed object fields for a USAF target for different relative amplitudes of second-order transverse mode excitation. As noted in the previous section, we estimate the energy of the second order mode in our experiments to be $\sim 2\%$ of the fundamental mode energy. Importantly, we note that the contributions of the higher-order modes is not a

noise-like or background contribution, but a coherent term that contains imaging information. As such, it may be used for improved imaging by digitally taking into account the distribution of the reference field in the higher modes: $E_{R,m}^{(j)}(x, y)$, and the difference in object illumination. This may be useful to improve the imaging NA and potentially the FoV, but will require a more involved reconstruction scheme, to simultaneously determine the object pattern and the modal excitation complex amplitudes.

Fig S6: Simulated effect of higher-order fiber modes on object reconstruction. Numerical investigation of the reconstructed object field when the relative delay between the illumination- and reference arms matches the object distance (i.e. case (ii) of Fig.S5). (A) Assuming only the fundamental mode LP_{01} is excited in both reference and illumination arms. (B) Assuming 2% of the illumination energy excites the second (LP_{11}) fiber mode in both arms, as in our experiments (see Fig.S5). (C) Displays the simulated contribution of the second mode (LP_{11}), to the hologram. (D,E,F) are zoom-in on the dashed boxes in (A,B,C).

Page 5, line 196, correct “reflectivity” twice in the same line

We have corrected this point in the revised manuscript

13. Fig 3J, why is the minimum distance of the object from mirror 60um or so? Is it the limitation of the system or of the measurement method? Also, it looks like the method is more noisy for smaller z_0 . At least that is my impression from Fig S3 but I might be wrong. I don't see any reason why that would happen. Can you please comment?

The minimal working distance of the presented approach is limited by the coherence length of the laser used and the reflectivity of the object, as it is required to separate the object reflection from the reflection of the partial mirror. The effect of the finite coherence length can be observed in Fig.2C. We have mentioned this point in the Methods section of the original manuscript:

“A shorter coherence-length source would be beneficial to both improve the axial-sectioning resolution and reduce the minimal working distance. “

This is also the reason for the added noise in the reconstruction of the object in the shortest distance in Fig.S3

To address this in the revised manuscript, we have added the following line to the main manuscript, after eq (2):

The object distance can also be found from the low coherence holograms by scanning the time delay, τ , and plotting the total energy of the reconstructed field at each time delay (**Error! Reference source not found.C**). The minimal working distance is thus a function of the coherence-length of the laser used, and the object reflectivity.

14. I find Fig 6B lacking information. I know this might be a bit too much to ask but I think a much nicer demonstration would be to make a sample with defined etched steps and then do the stitching on this. For me, that would be a much better proof of depth sensing. I actually think Fig A is enough. If B is there just to push some biology in, I would say remove it, because it has little meaning in its current form. I also think that axial sectioning resolution can be discussed at this point, including limitations and that choosing a laser with shorter coherence length might improve axial sectioning.

We thank the referee for these remarks and for the honest opinion. Our motivation for originally including Fig.6B was to demonstrate depth-resolved imaging of a large FoV via mosaicking/stitching. To address this point we have replaced figure 6B with a result obtained from a resolution target inclined at an angle with respect to the distal fiber end. The revised figure and figure caption is:

Figure 6: Imaging three-dimensional objects. (A) A 3D image of a target composed of two stacked resolution targets with a spacing of $\sim 300\mu\text{m}$ reconstructed by super-posing two reconstructed images acquired with two appropriate time delays, $\tau = 2z_0/c$, with $z_0 = 1030\mu\text{m}$ (pink) and $z_0 = 1320\mu\text{m}$ (yellow). (B) A tilted USAF reflectivity target, reconstructed by stitching 20×20 sub-images, each with the fiber field of view given by the dashed square. The varying depth of each sub-image is retrieved from the time-delay scan. The target was placed at an angle w.r.t the distal tip to demonstrate 3D imaging capabilities. Scale bars: A - $50\mu\text{m}$ B - $500\mu\text{m}$

Following this question and Q4 of Referee 1, we have provided additional more accurate information on the axial-sectioning in the answer to Q1 above.

15. *Supplementary: Page 3, Line 70 correct “reflectivity”*

We have corrected this point in the revised manuscript

16. *Supplementary: Page 4, Line 105 correct “10 nm”. nm as subscript*

We have corrected this point in the revised manuscript

17. *Fig S1: There is some spurious s under figure caption*

We have corrected this point in the revised manuscript

18. *Fig S3C: the $z_0 = 90\mu\text{m}$ really looks noisier. Any reason why?*

As we noted in our reply to question 14 above, the reason for the added noise in smaller object distances is the strong reflection from the partial mirror, which is within the coherence gating window. This minimal distance is set by the coherence function of the laser, which is characterized in the new Supplementary Figure S7.

19. *Fig S4: It was not clear to me why this was included. Maybe a short discussion why imaging using these fibers is important would suffice.*

The reason for including these fibers is that they are widespread in many endoscopic applications and previous works in the field, mainly due to their higher fill-factor, reduced dimensions, and reduced cost. They also include cores that are positioned on a non-periodic (disordered) grid, which may be an advantage for compressed sensing algorithms that may improve reconstruction quality of high spatial frequencies. However, the higher fill-factor of these fibers comes together with increased cross-talk between the cores. Since this may be an important experimental factor for the performance of our system, we saw a need in demonstrating our approach also with these common commercial fibers.

To note these points we have added the following line was to the revised manuscript discussion:

“FiDHo can also be employed with MCFs having disordered cores positions, as is demonstrated in supplementary Fig S4. The random spatial sampling of such MCFs may allow improved reconstruction via compressed-sensing reconstruction algorithms [36] “

REVIEWERS' COMMENTS

Reviewer #1 (Remarks to the Author):

The questions were explained comprehensively. Actually, the image quality is crucial for several applications. However, the authors have addressed, for example, the approach exploiting benefits of the used ultrashort laser pulses.

The revised paper "Real-time holographic lensless micro-endoscopy through flexible fibers via Fiber Bundle Distal Holography (FiDHo)" was improved significantly. It was a pleasure to read the excellent-written paper and the detailed comments on the reviews.

The paper is ready for publication.

Reviewer #2 (Remarks to the Author):

The authors have adequately addressed my comments and concerns.

I support the publication of this work in Nat. Comm.

Reviewer #3 (Remarks to the Author):

The authors have addressed all my concerns and answered all my questions. The added information and material are well presented and go beyond what I expected, especially the new miniaturisation section was revealing and really well explored. I would like to thank the authors for their hard work and I am happy to recommend the manuscript for publication in Nature Communications.

Martin Ploschner